# Dynamic Sparse Training versus Dense Training: The Unexpected Winner in Image Corruption Robustness

**Boqian Wu**[1,2,*]**, Qiao Xiao**[3,*]**, Shunxin Wang**[1]**, Nicola Strisciuglio**[1]**, Mykola Pechenizkiy**[3]**,**
**Maurice van Keulen**[1]**, Decebal Constantin Mocanu**[2,3]**, Elena Mocanu**[1]
[1] University of Twente, [2] University of Luxembourg, [3] Eindhoven University of Technology
`{b.wu,s.wang-2,n.strisciuglio,m.vankeulen,e.mocanu}@utwente.nl`
`{q.xiao,m.pechenizkiy}@tue.nl, {decebal.mocanu}@uni.lu`

## Abstract

It is generally perceived that Dynamic Sparse Training opens the door to a new era of scalability and efficiency for artificial neural networks at, perhaps, some costs in accuracy performance for the classification task. At the same time, Dense Training is widely accepted as being the "de facto" approach to train artificial neural networks if one would like to maximize their robustness against image corruption. In this paper, we question this general practice. Consequently, *we claim that*, contrary to what is commonly thought, the *Dynamic Sparse Training* methods can consistently *outperform Dense Training* in terms of *robustness accuracy*, particularly if the efficiency aspect is not considered as a main objective (i.e., sparsity levels between 10% and up to 50%), without adding (or even reducing) resource cost. We validate our claim on two types of data, images and videos, using several traditional and modern deep learning architectures for computer vision and three widely studied Dynamic Sparse Training algorithms. Our findings reveal a new yet-unknown benefit of Dynamic Sparse Training and open new possibilities in improving deep learning robustness beyond the current state of the art.

## 1 Introduction

If one would like to maximize as much as possible the robustness of a deep learning model on noisy more realistic data (e.g., image corruption, out of distribution), what would be the typical approach? Would one train directly a dense neural network (referred further as Dense Training) or a sparse neural network (referred further as Sparse Training)? Of course, there are many techniques to enhance the model robustness, but still they would be applied on top of a basic optimization process which usually requires training an artificial neural network. In order to give an answer to the previously posed question, we would need to quantify the research and practitioner community perception, which is a hard problem in itself. Yet, given the large body of literature on the topic, it is safely to assume that the usual answer would be "Yes, Dense Training is the typical solution to maximize the model robustness.". Nevertheless, the reader can answer to this question by themselves, while further in this manuscript we would like to bring more clarity and some new perspectives on this topic.

With respect to Dense Training, recent studies have demonstrated that over-parameterized neural networks, can achieve impressive generalizability on test data even without explicit regularizers Zhang et al. (2021). This phenomenon is partly explained by the implicit regularization[1] induced by the optimization algorithm used during training, such as stochastic gradient descent (SGD) Zhang et al. (2021); Keskar et al. (2017); Li & Liang (2018). Despite ongoing efforts to understand the robustness behavior of deep neural network models, researchers always favor larger models, trained

---

[*]Equal contribution.

[1]In older neural network literature, regularization referred specifically to *a penalty term added to the loss function to prevent overfitting*. Recently, as in Goodfellow et al. (2016), regularization is broadly defined as "*any modification to a learning algorithm intended to reduce test error without increasing training error.*"

densely, due to their perceived robustness to unexpected variations in input data, such as additional noise, contrast changes, and weather conditions, to a certain extent Hendrycks & Dietterich (2019); Hendrycks et al. (2021). However, larger over-parameterized models do not necessarily gain more robustness Algan & Ulusoy (2021); Wang et al. (2023).

With respect to sparse training, there are static sparse training Liu & Wang (2023) and Dynamic Sparse Training (DST) Mocanu et al. (2018); Bellec et al. (2018); Mostafa & Wang (2019); Evci et al. (2020); Yuan et al. (2021); Liu et al. (2021a), which maintain just a fraction of the parameters' budget throughout the entire training process. Static sparse training involves sparsifying the network before training and keeps the topology fixed during training. DST, a recent and efficient sparse-to-sparse training approach, starts with a sparse neural network and simultaneously optimizes both its weight values and connectivity. This optimization process is known as dynamic sparsity. Consequently, it is mainly seen as an approach that can maximize deep learning scalability and efficiency, at least in theory, as in practice, it is a difficult research problem to fully utilize sparsity during training Nowak et al. (2023). Nevertheless, it is often overlooked that DST frequently achieves performance comparable to, or even surpassing, that of its dense or static sparse counterparts Chen et al. (2022b); Xiao et al. (2022); Graesser et al. (2022); Sokar et al. (2022); Atashgahi et al. (2022); Liu et al. (2023); Sokar et al. (2023).

DST shares many similarities with dense training, including its adaptability to a range of learning paradigms, such as continual learning Sokar et al. (2023) and reinforcement learning Graesser et al. (2022), as well as tasks like feature selection Atashgahi et al. (2022), time series classification Xiao et al. (2022). It is compatible with both convolutional and transformer-based architectures and uses core techniques such as dropout, weight decay, and standard optimizers Dey et al. (2024). Despite these parallels, DST utilizes hard sparsity—on top of the soft sparsity introduced by weight decay or other regularization methods—which provides additional regularization. This naturally leads to an intriguing question: *Is somehow DST implicitly improving model robustness, and no one knew about this yet?*

Although several studies have investigated the impact of pruning model from Dense Training, lottery ticket hypothesis methods Frankle & Carbin (2019); Frankle et al. (2020); Renda et al. (2020) on image corruption robustness Liebenwein et al. (2021); Diffenderfer et al. (2021) and adversarial robustness Liao et al. (2022), other research shows that static sparse networks, when increased width and depth to maintain capacity, can also improve robustness Timpl et al. (2022). However, a closer look at current DST research reveals a literature gap: the overlooked evaluation of DST models under various unexpected image corruptions. In an attempt to respond to this, we examined one of the classical DST algorithms, Sparse Evolutionary Training (SET) Mocanu et al. (2018), as illustrated in Figure 1 (Left). We find that SET (with a sparsity ratio of 0.5) can markedly improve model robustness against 18 out of 19 image corruptions, compared to Dense Training. Intuitively, adding regularization (e.g., weight decay or dropout) during training is commonly believed to help improve the generalization of neural networks. In light of the strong performance of the DST method (e.g., SET) against common corruptions , we propose the Dynamic Sparsity Corruption Robustness (DSCR) Hypothesis.

**DSCR Hypothesis**. *Dynamic Sparse Training at low sparsity levels helps to improve model robustness against image corruption in comparison with Dense Training.*

We summarize our main contributions as follows:

- This study questions the typical approach of using Dense Training to maximize model robustness and proposes Dynamic Sparse Training as a more capable alternative.

- Formally, we propose the DSCR hypothesis which is validated in a wide range of corruption scenarios (i.e., corruption type and severity), using different types of data (i.e., image and video), DST algorithms (i.e., SET, RigL, MEST and GraNet) and deep learning architectures (i.e., VGG, ResNet, EfficientNet, DeiT and 3D ConvNet) (Section 4).

- We give insights, from both spatial and spectral perspectives, on the underlying factors that explain the superior performance of DST. Our analysis shows that the dynamic sparsity process in DST acts as an implicit regularization mechanism, enabling the model to automatically focus on more important features (e.g., low-frequency components) while reducing its attention to less important ones (e.g., high-frequency components) (Section 5).

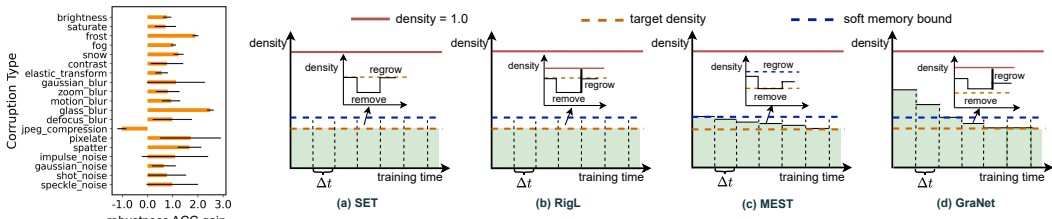

Figure 1: (Left) Robustness accuracy gain (%) in a sparsified ResNet34 trained with SET (sparsity ratio=0.5) compared to its dense counterpart, tested on CIFAR100-C with various corruption types shown on the Y-axis. Positive values indicate better performances of SET method. (Right) conceptual representation of model density during training across different DST algorithms: (a) SET, (b) RigL (Note: In the regrow step, RigL requires full gradient calculations.), (c) MEST and (d) GraNet.

## 2 RELATED WORK

### 2.1 RELEVANT METHODS FOR CORRUPTION ROBUSTNESS

Deep neural networks (DNNs) are susceptible to unseen changes in their inputs, a scenario commonly encountered in practical applications, such as common corruptions like Gaussian noise, defocus blur, etc Hendrycks & Dietterich (2019). To enhance robustness against these common corruptions, many methods have been developed, primarily focusing on data augmentation, for example, AugMix Hendrycks* et al. (2020), AugMax Wang et al. (2021), PRIME Modas et al. (2022). These approaches aim to reduce the distribution gap between the training and testing datasets. Alternatively, other strategies to enhance corruption robustness concentrate on developing resilient learning approaches, such as contrastive learning Chen et al. (2020), or on designing specialized network components, like the push-pull layer Strisciuglio et al. (2020). Recent research has revealed that LTH-based sparse models can provide enhanced robustness to common corruptions compared to their denser counterparts Diffenderfer et al. (2021); Chen et al. (2022a). Additionally, Bair et al. (2024) introduces an adaptive sharpness-aware pruning method, which regularizes the model towards flatter regions, thereby improving robustness against image corruption. However, these methods are typically built on dense models either at the start or throughout training, often overlooking the aspect of resource-aware training.

### 2.2 DYNAMIC SPARSE TRAINING

Dynamic Sparse Training (DST) Mocanu et al. (2018); Bellec et al. (2018); Mostafa & Wang (2019); Evci et al. (2020); Chen et al. (2021); Yuan et al. (2021); Wu et al. (2025), as a sparse-to-sparse training paradigm, starts from scratch and maintains partial parameters throughout training, providing a promising approach to significantly improve training and inference efficiency while still preserving accuracy. Recent research has focused on deciphering the mechanisms of DST Liu et al. (2021c); Evci et al. (2022; 2019), and has broadened its application in various domains, such as continuous learning Sokar et al. (2023), reinforcement learning Graesser et al. (2022); Tan et al. (2023), feature selection Sokar et al. (2022); Atashgahi et al. (2022), time series classification Xiao et al. (2022), network architecture design Liu et al. (2023), and vision transformers Chen et al. (2021).

Moreover, recent studies have concentrated on developing improved DST algorithms Dettmers & Zettlemoyer (2019); Mostafa & Wang (2019); Yuan et al. (2021); Jayakumar et al. (2020); Liu et al. (2021a). For instance, Dettmers & Zettlemoyer (2019) introduced a technique to regrow weights based on the momentum of existing weights, while Yuan et al. (2021) proposed a method that utilizes both weight and gradient information for pruning. More recently, it has been demonstrated that DST can enhance the performance of DNNs on biased data Zhao et al. (2023) and adversarial samples Chen et al. (2022b). However, the majority of DST research relies on pristine benchmark datasets, leaving the robustness of DST models against image corruption relatively unexplored. This paper aims to systematically investigate the robustness behavior of DST when exposed to various image corruptions.

## 3 PRELIMINARIES

### 3.1 PROBLEM DEFINITION

Given $n_{\text{tr}}$ i.i.d. training samples $\{(\boldsymbol{x}_i^{\text{tr}}, y_i^{\text{tr}})\}_{i=1}^{n_{\text{tr}}}$ and $n_{\text{te}}$ test samples $\{(\boldsymbol{x}_i^{\text{te}}, y_i^{\text{te}})\}_{i=1}^{n_{\text{te}}}$, we aim to evaluate the model's robustness in scenarios where the test inputs are corrupted relative to the benchmarked training datasets, (i.e., $p_{\text{tr}}(\boldsymbol{x}) \neq p_{\text{te}}(\boldsymbol{x})$) while the conditional input-label relationship remains consistent (i.e., $p_{\text{tr}}(y \mid \boldsymbol{x}) = p_{\text{te}}(y \mid \boldsymbol{x})$). To achieve this, we utilize standard image benchmark datasets such as CIFAR10 Krizhevsky et al. (2009), CIFAR100 Krizhevsky et al. (2009), TinyImageNet Le & Yang (2015) and ImageNet Deng et al. (2009) for training. Then, we evaluate the performance (robustness accuracy) of the models on the corresponding corruption benchmark datasets, including CIFAR10-C, CIFAR100-C, TinyImageNet-C and ImageNet- C/C̄/3DCC. More details of datasets can be found in Appendix A.2.

### 3.2 DYNAMIC SPARSE TRAINING

Consider a dense neural network $f_{\boldsymbol{\theta}}$, the sparsification of $f_{\boldsymbol{\theta}}$ can be formulated as: $f_{\boldsymbol{\theta} \odot \mathbf{M}} \mapsto f_{\boldsymbol{\theta}'}$, where $\mathbf{M}$ is a set of binary masks and $\odot$ denotes element-wise multiplication. The sparsity ratio $S$ of the resulting subnetwork, $f_{\boldsymbol{\theta}'}$, is defined as $S = 1 - \frac{\|\boldsymbol{\theta}'\|_0}{\|\boldsymbol{\theta}\|_0}$, where $\|\cdot\|_0$ represents the $L_0$ norm, counting the number of non-zero elements. The target memory budget (or density) ratio $b$ is given by $\frac{\|\boldsymbol{\theta}'\|_0}{\|\boldsymbol{\theta}\|_0}$. The topology of the sparse network is dynamically updated during training every $\Delta t$ iterations through a parameters remove-regrow process. Specifically, connections with low importance are removed, and new connections are regrown based on criteria like gradient information Evci et al. (2020) or randomly Mocanu et al. (2018). Dynamic sparse models are the resulting models from DST. In our study, we explore various DST approaches as follows:

- DST with **fixed memory budget in forward**: To the best of our knowledge, SET is pioneering work in DST. SET follows a magnitude-based removal and random regrow strategy, maintaining a fixed the memory budget, that is $\frac{\|\boldsymbol{\theta}'\|_0}{\|\boldsymbol{\theta}\|_0} \equiv b$, throughout the entire training process, as shown in Figure 1 (right (a)). RigL Evci et al. (2020) regrows weights based on the gradient information of all weights, while maintaining a constant memory budget during the forward phase of training, as shown in Figure 1 (right (b)). In both approaches, the mask $\mathbf{M}$ is updated every $\Delta t$ iterations.

- DST with **soft memory budget in forward**: Memory-Economic Sparse Training (MEST) Yuan et al. (2021), with Soft Memory Bound Elastic Mutation, starts with a sparse model having a memory budget of $\frac{\|\boldsymbol{\theta}'\|_0}{\|\boldsymbol{\theta}\|_0} = b + b_s$, where $b_s > 0$ represents the soft memory bound that decays during training. Subsequently, at intervals of $\Delta t$, weights are removed based on magnitude and gradient information, reducing the memory budget to $b$, and then regrown it to $b + b_s$, as illustrated in Figure 1 (right (c)). The soft memory constraint $b_s$, gradually decays to zero, allowing the model to reach the target memory budget $b$.

- DST with **flexible memory budget in forward**: GraNet Liu et al. (2021a) starts with a denser neural network (i.e., $\frac{\|\boldsymbol{\theta}'\|_0}{\|\boldsymbol{\theta}\|_0} < 1$). During training, at intervals of $\Delta t$, it removes weights based on magnitude and regrows connections using gradient information, similar to RigL. The main difference is that fewer connections are regrown than those removed. This process dynamically updates the mask $\mathbf{M}$ throughout training, gradually reaching the target memory budget ratio $b$ by the end, as shown in Figure 1 (right (d)).

In this paper, for simplicity in analysis, we use a unified magnitude-based removal approach and implement either random $((\cdot)_r)$ or gradient-based (i.e., re-growing the weights with the highest magnitude gradients)$((\cdot)_g)$ regrow strategy across different DST methods. Unlike their original papers Yuan et al. (2021); Liu et al. (2021a), where MEST utilized random regrow and GraNet employed gradient-based regrow, we explore both strategies for each model. Thus, for random regrow DST, our analysis includes SET, $\text{MEST}_r$, and $\text{GraNet}_r$, while for gradient-based regrow DST, we examine RigL, $\text{MEST}_g$, and $\text{GraNet}_g$. This approach allows us to consistently evaluate the robustness of DST models under varying memory budget constraints, such as fixed, soft, or flexible, during training.

# 4 CAN DST IMPROVE ROBUSTNESS AGAINST IMAGE CORRUPTION?

In this section, we aim to carry out extensive experiments to verify the DSCR Hypothesis across a wide variety of DST methods, models and data. To this end, we conduct a comprehensive set of experiments to evaluate the robustness accuracy of DST models with different sparsity ratios in the presence of image corruption.

To comprehensively validate the DSCR Hypothesis, our study embarks on an extensive analysis evaluating the robustness behavior of sparsely trained models. We train using several representative DST algorithms, including SET, Rigging the Lottery (RigL) Evci et al. (2020), Memory-Economic Sparse Training (MEST) Yuan et al. (2021), and Gradual Pruning with zero-cost Neuroregeneration (GraNet) Liu et al. (2021a), across a wide range of model architectures (i.e., VGG Simonyan & Zisserman (2015), 2D/3D ResNet He et al. (2016), EfficientNet Tan & Le (2019), DeiT Touvron et al. (2021) and Two-Stream Inflated 3D ConvNet (I3D) Carreira & Zisserman (2017)).

Our analysis spans a range of image corruption benchmarks—CIFAR10-C Hendrycks & Dietterich (2019), CIFAR100-C Hendrycks & Dietterich (2019), TinyImageNet-C Hendrycks & Dietterich (2019), ImageNet-C Hendrycks & Dietterich (2019), ImageNet-C̄ Mintun et al. (2021), ImageNet-3DCC Kar et al. (2022) and corrupted UCF101 Soomro et al. (2012) —covering 19 types of image corruptions for CIFAR10/100-C, 15 types for TinyImageNet-C and ImageNet-C, 10 types for ImageNet-C̄ and 11 types for ImageNet-3DCC, with each corruption tested across five severity levels. The corrupted UCF101 dataset includes 16 corruption types with four severity levels. This analysis provides a comprehensive perspective on the robustness evaluation of sparsely trained models. Detailed dataset descriptions and implementation setup can be found in Appendix A.1 and A.3.1.

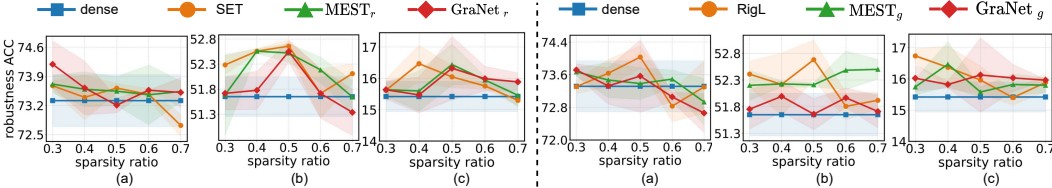

Figure 2: Robustness accuracy (%) for (a) VGG16 on CIFAR10-C, (b) ResNet34 on CIFAR100-C and (c) EfficientNet-B0 on TinyImageNet-C, comparing different DST algorithms with dense training. Left: Results for DST algorithms with random regrow strategy, including SET, $\text{MEST}_r$, and $\text{GraNet}_r$. Right: Results for DST algorithms with gradient-based regrow strategy, such as RigL, $\text{MEST}_g$, and $\text{GraNet}_g$.

## 4.1 OVERALL PERFORMANCE

Figure 2 showcases the average robust accuracy across various types of corruption, with severity levels ranging from 1 to 5, on the CIFAR10/100-C and TinyImageNet-C datasets. We can find that DST models outperform their Dense training counterparts (as indicated by the blue line in Figure 2) in terms of robustness accuracy across several sparsity ratios. This trend is evident in both the random regrow (Figure 2, left) and gradient-based regrow (Figure 2, right) strategies. Specifically, for CIFAR100-C and TinyImageNet- C, DST models with sparsity ratios ranging from 0.3 to 0.7 generally show encouraging improvements in robustness. For CIFAR10-C, at certain sparsity ratios, such as 0.4, the overall DST methods achieve decent robust performance against image corruptions. Moreover, based on our experimental setup, at least 40% of both training computational and memory costs can be saved while simultaneously enhancing corruption robustness. Details on FLOPs and parameter counts are available in the Appendix A.3.2. This finding substantiates our DSCR Hypothesis across different DST methods, varying sparsity ratios, and architectures (i.e., VGG, ResNet, and EfficientNet).

## 4.2 PERFORMANCE ON CORRUPTION TYPES

We further evaluate the DST methods across various corruption types with different severities. Figure 3 provides detailed relative robustness gains [2] for each type of corruption in ImageNet-C and

---

[2]Defined as $(\text{Acc}_{s,sl} - \text{Acc}_{d,sl})/\text{Acc}_{d,sl}$, where $\text{Acc}_{s,sl}$ and $\text{Acc}_{d,sl}$ represent the accuracies of models trained using DST and dense training, respectively, at a severity level of $sl$.

ImageNet-$\bar{\text{C}}$ Mintun et al. (2021), compared to their dense model counterparts. It is worth mentioning that the X-axis in Figure 3 represents different types of corruption, which are ordered according to the amount of high-frequency information they contain, as in Saikia et al. (2021). Intriguingly, on ImageNet-C and ImageNet-$\bar{\text{C}}$ [3], DST models show significant potential in enhancing robustness against specific corruptions, such as impulse, Gaussian, and shot noise on ImageNet-C, as well as brownish, Perlin, plasma, and blue noise on ImageNet-$\bar{\text{C}}$. It is worth noting that these corruptions are typically characterized by relatively high-frequency information. This finding is consistent on CIFAR10-C, CIFAR100-C, and TinyImageNet-C, and due to page limitations, the results for these datasets are detailed in Appendix A.4.

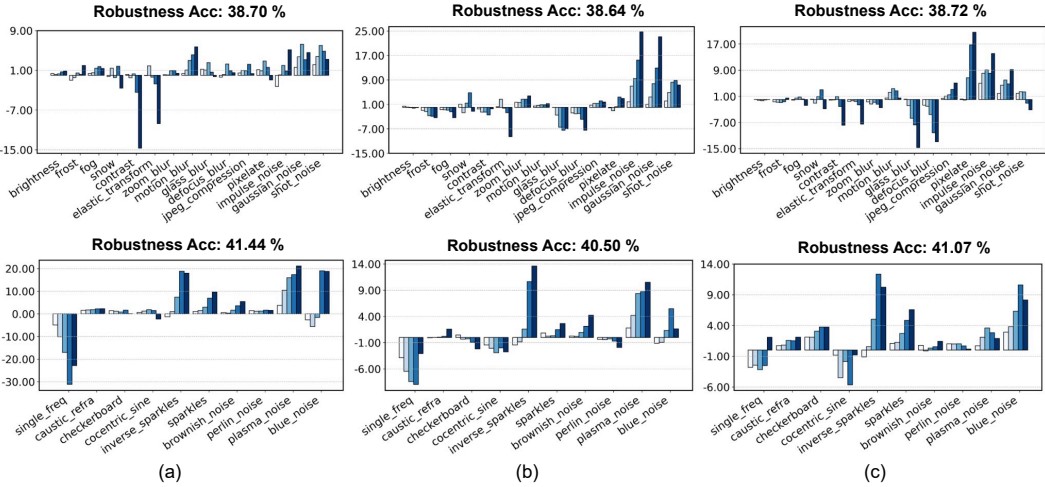

(a)        (b)        (c)

Figure 3: Relative robustness accuracy gain (%) on ImageNet-C (top) and ImageNet-$\bar{\text{C}}$ (bottom) for ResNet50, trained using DST with gradient-based strategies (i.e., (a) RigL, (b) MEST$_g$, (c) GraNet$_g$) at a sparsity ratio of 0.1, compared to a dense baseline (which has a mean robustness accuracy of $38.38\%$ on ImageNet-C and $40.38\%$ on ImageNet-$\bar{\text{C}}$). Positive values reflect better performance by the sparse models compared to the dense baseline. The five bars, ranging from light to dark shades, represent corruption severity levels from 1 to 5 for each type of corruption. The title indicates the mean robustness accuracy for each corresponding DST method.

Additionally, we observe that DST models exhibit a more pronounced advantage in robustness over Dense Training models under high severity levels, particularly with noise corruptions. Specifically, as shown in Figure 3 (top (b)), the robustness performance of MEST$_g$ at severity level 5 for impulse noise and Gaussian noise demonstrates nearly a 25% improvement. This finding is also evident in CIFAR10-C, CIFAR100-C, and TinyImageNet-C; more details can be found in Appendix A.4. This observation highlights the potential of models trained with DST to tackle high-frequency corruptions, especially in harsher scenarios, offering the potential to enhance DNNs' robustness in complex corruption-prone environments.

### 4.3 ROBUSTNESS OF DST MODELS: IS TRUE FOR OTHER DATASETS, TASKS, AND ARCHITECTURES?

Herein, we evaluate the robustness of DST models using a more realistic corruption dataset (e.g., ImageNet-3DCC Kar et al. (2022)), and across different tasks (e.g., video classification), with modern architectures (transformer-based, e.g., DeiT-base Touvron et al. (2021)).

**Evaluation on ImageNet-3DCC.** We evaluate DST models on the ImageNet-3DCC, a challenging dataset with realistic corruptions. The ImageNet-3DCC dataset introduces corruptions aligned with 3D scene geometry, simulating real-world shifts including camera motion, weather conditions, oc-

---

[3]Our experiments primarily focus on DST with gradient-based regrow strategy when training on ImageNet. As our implementation is based on PyTorch, this regrow approach tends to achieve strong performance on ImageNet more quickly with this framework Liu et al. (2021a). However, it is worth noting that the random-based approach has also shown significant improvements when implemented in JAX Lee et al. (2024), which can be future exploration.

clusions, depth of field, and lighting variations. The improved average test accuracy on ImageNet-3DCC, as shown in Table 1, indicates that DST models excel not only at handling synthetic distortions, as discussed in Section 4.1 and Section 4.2, but also at adapting to practical, real-world corruptions.

**Evaluation on video classification.** We extended our exploration to video data using the UCF101 dataset Soomro et al. (2012), an action recognition dataset containing 101 categories and approximately 13K videos. Following the settings described in Schiappa et al. (2023), we compared the performance of dense training and DST at a sparsity level of 0.1 on two archi-

Table 1: The robustness accuracy (%) of dense and DST models on ImageNet-3DCC and UFC101 datasets.

| Dataset | Model | Dense | RigL | MEST$_g$ | GraNet$_g$ |
|---|---|---|---|---|---|
| ImageNet-3DCC | ResNet50 | 43.62 | **44.08** | 43.87 | **44.46** |
| UCF101 | 3D ResNet50 | 51.14 | **52.71** | 53.00 | **53.57** |
| | I3D | 53.58 | **55.00** | 56.30 | **55.21** |

tectures: 3D ResNet50 and Two-Stream Inflated 3D ConvNet (I3D) Carreira & Zisserman (2017). Further implementation details are provided in Appendix A.3.1. Both models were trained from scratch for 100 epochs on UCF101. Our analysis, based on a test set featuring 16 corruptions across severity levels 1 to 4, as shown in Table 1, revealed that DST models consistently achieved superior robust accuracy compared to their dense counterparts, even within the context of video data.

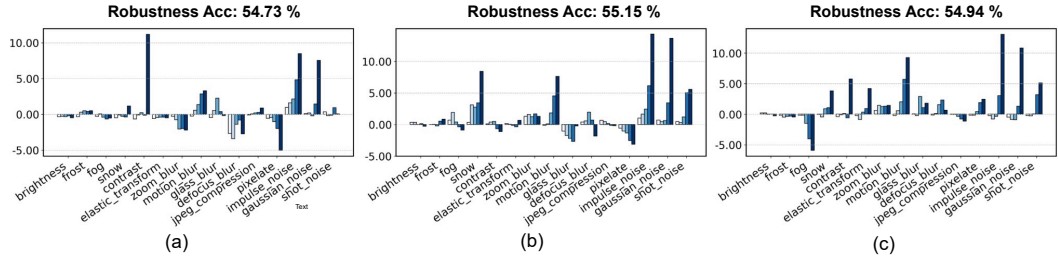

Figure 4: Relative robustness accuracy gain (%) on ImageNet-C for a DeiT-base, dynamically trained with gradient-based methods (i.e., (a) RigL, (b) MEST$_g$, (c) GraNet$_g$) at a sparsity ratio of 0.1, compared to a dense baseline (with a robustness accuracy of 54.68% on ImageNet-C). Positive values indicate superior performance by the sparse models.

**Evaluation on transformer-based architecture.** We evaluate the relative robustness accuracy gain of various sparse and dense DeiT-base models on ImageNet-C across different corruption types, as shown in Figure 4. Following the experimental settings in Chen et al. (2021), we conducted training from scratch over 200 epochs. The results show that DST models generally outperform their dense counterparts, with a particularly better advantage in handling noise corruptions. This finding, consistent with our observations in CNNs discussed in previous sections, suggests that the robustness benefits of DST models are applicable to alternative architectures, including transformer-based model.

In Appendix A.9, we compare the robustness of dense trained and DST models on an image segmentation task and find that DST models are robust not only in image and video classification but also in image segmentation. In summary, our empirical findings indicate that *DST methods can surpass Dense Training models in robustness performance across various datasets at specific sparsity ratios, without (or even reduce) extra resource cost* [4]. Furthermore, DST models tend to perform better against different types of corruption, particularly those with more high-frequency information. The mystery behind these observations will be further explored in the following sections.

## 5 WHAT IS BEHIND THE CORRUPTION ROBUSTNESS GAIN IN DST?

Building on the previous empirical results, where DST showed improved robustness against common corruptions, we are motivated to explore the inner reasons for the robustness of DST models. Our focus is on analyzing how DST and Dense Training models utilize their parameters in the spatial domain and process the input in the spectral domain.

---

[4]We use a binary masking operation to simulate the sparisty, the computation cost of binary masking operation is negligible or even reduced when the hardware support sparse matrix product.

## 5.1 ANALYSIS THROUGH SPATIAL DOMAIN

Traditionally, the constraints on weight values, such as $L_2$ regularization, during dense training have been an effective method to mitigate the problem of overfitting. In contrast, DST imposes sparsity constraints, enabling the selective removal of parameters. In this section, we aim to intuitively investigate how DST affects the location of non-zero weights. To achieve this, we performed a visualization analysis to pinpoint the locations of sparse parameters for DST models. Figure 5 showcases the non-zero weight counts and the sum of weight magnitudes (i.e., absolute values) for each kernel in a specific layer.

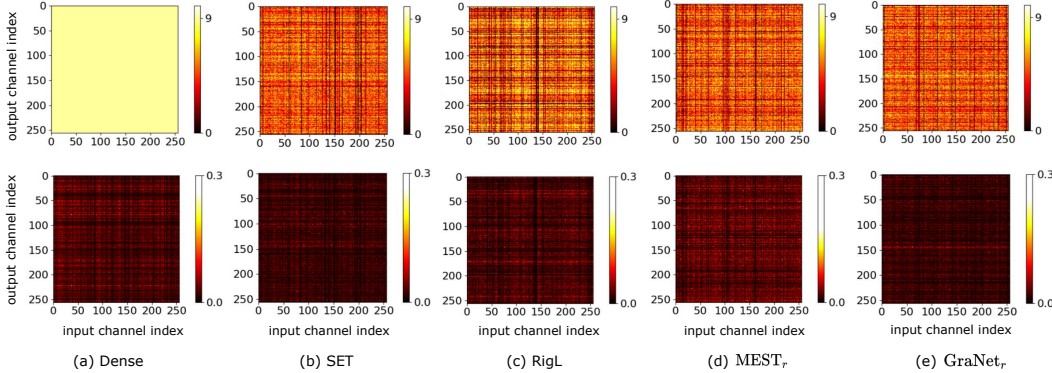

(a) Dense (b) SET (c) RigL (d) MEST$_r$ (e) GraNet$_r$

Figure 5: Visualization of the non-zero weight count (1st row) and the sum of weight magnitudes (2nd row) within a $3\times3$ kernel from the #layer3.3.conv2 of ResNet34 after training on the CIFAR100 using dense training or different DST methods (with a sparsity ratio of $0.5$). Each value in the figure is derived from calculations within the kernel, with lighter colors indicating larger values.

Interestingly, the heatmaps in Figure 5 show dark vertical and horizontal lines, indicating a concentration of either sparsified weights or weights with smaller magnitudes in the filters corresponding to specific input or output channels (features). While both dense and DST models exhibit such a concentration phenomenon, in the case of the dense model, this concentration is solely due to the adjustment of weight magnitudes. These magnitudes tend to converge near, but not exactly at, zero, as indicated by the consistently non-zero weight count of 9 in Figure 5 (1st row (a)). In contrast, the dynamic sparse model attributes this phenomenon to both the adjustment of weight magnitudes and the allocation of sparsified weights, as shown in Figure 5 (1st row (b, c, d, e)), where the non-zero weight count approaches zero. The findings are consistently evident in other layers, particularly in deeper layers, and across different models. More visualizations can be found in the Appendix A.5. Compared to dense training, we suggest that dynamic sparsity in DST introduces a form of implicit regularization. This leads to the prominence of sparsified weights on particular filters, indicating a **reduced connectivity to specific features that are determined to be less crucial**.

Furthermore, based on the empirical results in Section 4.2, we observed that DST models are surprisingly more effective at countering corruptions rich in high-frequency information compared to their dense training counterparts. We assume that the improved robustness of DST models against these corruptions stems from their inherent tendency to de-emphasize high-frequency features. This selective focus on specific frequency ranges enhances the network's robustness against high-frequency corruptions, thereby providing a distinct advantage in handling noise corruptions rich in high-frequency information. Further analysis will be provided in the following section.

## 5.2 ANALYSIS FROM SPECTRAL DOMAIN

Motivated by previous work Yin et al. (2019), which examines ImageNet accuracy after low-pass and high-pass filtering to understand how neural networks utilize frequency information for image recognition, and building on the observations and assumptions from the previous section, we will further explore dynamic sparse models from a spectral perspective.

In this section, we explore how the test accuracy of both DST and Dense Training models is affected by the removal of high and low-frequency components from the test data after training. In detail, the test images are first transformed into the Fourier spectrum, followed by a frequency attenuation

operation. The attenuation is applied in two forms: high-frequency attenuation $a_{high}$ and low-frequency attenuation $a_{low}$, each controlled by an attenuation radius $r$. A larger $r$ leads to more information being filtered out, as depicted in Figure 6.

We define $\mathrm{Acc}_{\mathtt{high}}(M, r)$ and $\mathrm{Acc}_{\mathtt{low}}(M, r)$ as the test accuracies of a model $M \in \mathcal{M}$, when tested on datasets processed with high-frequency attenuation $a_{high}$ and low-frequency attenuation $a_{low}$ respectively, using an attenuation radius $r$. Here, $\mathcal{M}$ represents the set of all considered models. The Radius-Accuracy (RA) curves for $a_{high}$ and $a_{low}$ illustrate the relationship between frequency attenuation range and test accuracy:

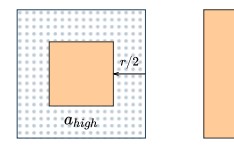

Fourier spectrum        Fourier spectrum

Figure 6: Illustration of high-frequency $a_{high}$, and low-frequency $a_{low}$ attenuation. Gray areas indicate the removed frequency components, with low frequencies near the center.

$$\mathcal{C}_{M,\mathtt{high}} := \{(r, \mathrm{Acc}_{\mathtt{high}}(M, r)) \mid M \in \mathcal{M}, r \in \mathcal{R}\}, \quad (1)$$

$$\mathcal{C}_{M,\mathtt{low}} := \{(r, \mathrm{Acc}_{\mathtt{low}}(M, r)) \mid M \in \mathcal{M}, r \in \mathcal{R}\}, \quad (2)$$

where $\mathcal{R}$ denotes the set of attenuation radius. Figure 7 (left) illustrates a toy example of ImageNet with attenuated high and low-frequency information at different attenuation radius, and provides the RA Curves $\mathcal{C}_{M,\mathtt{high}}$ and $\mathcal{C}_{M,\mathtt{low}}$ colored with orange and blue, respectively. In detail, with high-frequency attenuation, images appear blurred but remain recognizable to the human. In contrast, low-frequency attenuation leaves only the basic outlines of objects visible. Figure 7 (right) shows the detailed RA curves for both dense and dynamic sparse models across different datasets. These models are tested on the CIFAR100, TinyImageNet, and ImageNet test sets, each processed with high- and low-frequency attenuation at varying radius.

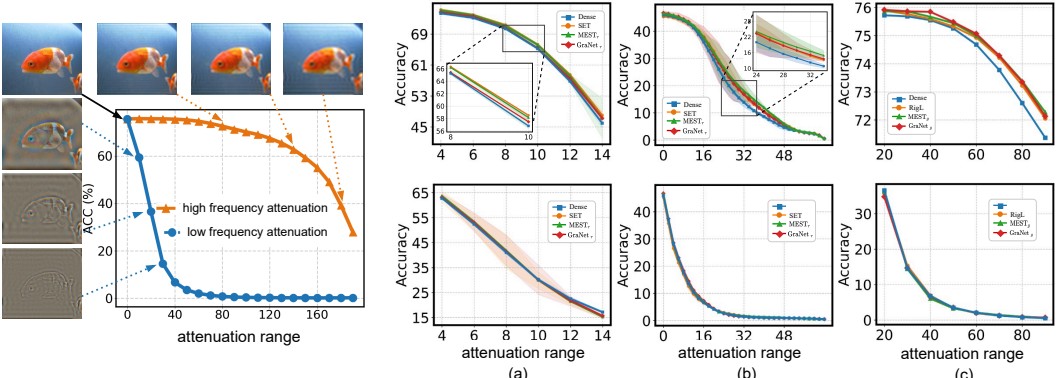

Figure 7: (Left) A toy example demonstrating the impact of high- and low-frequency attenuation on dense ResNet50 model using the ImageNet test set. (Right-top) High-frequency attenuation and (Right-bottom) Low-frequency attenuation performance: (a) ResNet34 (sparsity $0.5$) on CIFAR100, (b) EfficientNet-B0 (sparsity $0.5$) on TinyImageNet, and (c) ResNet50 (sparsity $0.1$) on ImageNet. The x-axis represents the attenuation radius, and the y-axis shows test accuracy.

**Observation 1: Equal attention for low-frequency information.** Given models $M \in \mathcal{M}$ and the set of attenuation radius $r \in \mathcal{R}$, it is observed that $\mathcal{C}_{M,\mathtt{low}}$ lie below the $\mathcal{C}_{M,\mathtt{high}}$. This reveals that attenuating low-frequency information significantly degrades classification accuracy, highlighting the crucial role of low-frequency information in image recognition tasks, consistent with findings in Yin et al. (2019). Under low-frequency attenuation $a_{low}$, both DST and Dense Training models exhibit similar performance, as shown in Figure 7 (right, 2nd row), suggesting that DST models allocate attention to low-frequency information in a manner comparable to their Dense Training counterparts.

**Observation 2: Less attention to high-frequency information:** When subjected to high-frequency attenuation $a_{high}$, DST models consistently outperform Dense Training models across all evaluated datasets, as shown in Figure 7 (right, 1st row). This suggests that high-frequency attenuation has less impact on DST models, indicating they are less sensitive to high-frequency information. This finding further supports the robustness of dynamic sparse models against common image corruptions rich in high-frequency content, as discussed in Section 5.1.

Based on the above observation, we conclude that DST introduces a form of implicit regularization, reducing the focus on high-frequency information. This selective attention in dynamic sparse models enhances their robustness, especially in handling corrupted images with high-frequency content, such as noise corruption.

Previous works Li et al. (2021); Grabinski et al. (2022) have pointed out that high-frequency components can cause frequency aliasing when downsampling operations (e.g., AveragePooling and Max-Pooling) are performed in the spatial domain of neural networks, as explained by the sampling theorem in information theory. Specifically, (Li et al., 2021) demonstrated that relying on low-frequency components of wavelets improves noise robustness, attributing this to the suppression of aliasing effects. Furthermore, (Grabinski et al., 2022; 2023) showed that removing high-frequency components to reduce frequency aliasing enhances the robustness of networks. These studies validate that reducing high-frequency components mitigates frequency aliasing and improves robustness, aligning with our findings on the relationship between DST models' tendency to pay less attention to high-frequency information and their improved robustness against common corruptions.

## 6 FURTHER ANALYSIS AND DISCUSSION

Our study questions the conventional approach of using "de facto" dense training for model robustness in favor of sparse training. Due to the page constraint, we also analyze the complementarity between DST and data augmentation (e.g., Mixup Zhang et al. (2018)) in Appendix A.7, and further explore DST's robustness with Out-Of-Domain (OOD) data and Grad-CAM Selvaraju et al. (2017) in Appendix A.8 and A.10, respectively. Furthermore, we extend and validate our DSCR hypothesis with more recent DST methods, including Cannistraci-Hebb soft training (CHTs) Zhang et al. (2024b;a), which employs more complex growth criteria and topology initialization, as detailed in Appendix A.6. In the future, it is worth moving beyond the basic fact revealed in this paper, i.e., vanilla DST outperforms vanilla Dense Training for model robustness, and to devise new DST methods for model robustness especially in the high sparsity regime, and to replace Dense Training with DST in the current state-of-the-art techniques for model robustness. Additionally, while we empirically analyze robustness behaviors between dense and sparse training and provide insights from both spatial and frequency domains, further work on the theoretical understanding of this phenomenon, along with a fine-grained exploration of different components of DST (e.g., topology initialization, regrowth methods and regrowth weight initialization), could provide deeper insights for the development of more robust and efficient networks. This would be a promising direction for future research.

## 7 CONCLUSION

This paper questioned the typical approach of using "de facto" Dense Training to maximize model robustness in deep learning, and revealed an unexpected finding. That is: "*Dynamic Sparse Training at low sparsity levels can train more robust models against image corruption than Dense Training.*". This finding has been confined formally in the DSCR Hypothesis and rigorously validated on nine scenarios (as summarized in Table 2 where a striking observation can be made - all DST algorithms studied outperform Dense Training in all scenarios). To understand what factors contribute to this compelling and intriguing DST behaviour, to the end, we present a deeper analyze. We show that

Table 2: Summary of the main results presenting the 9 scenarios studied in this paper. The table takes a snapshot at a particular sparsity level and presents the robustness accuracy (%) of the gradient-based regrow DST methods. The bold values represent the best performer between Dense Training and each DST method taken separately.

| Data type | Dataset | Model | Dense Training | Dynamic Sparse Training | | | |
|---|---|---|---|---|---|---|---|
| | | | | Sparsity | RigL | MEST$_g$ | GraNet$_g$ |
| images | CIFAR10-C | VGG16 | 73.32 | 0.4 | **73.64** | **73.47** | 73.32 |
| | CIFAR100-C | ResNet34 | 51.65 | 0.4 | **52.56** | **52.56** | **51.78** |
| | TinyImageNet-C | EfficientNet-B0 | 15.42 | 0.4 | **16.46** | **16.5** | **15.82** |
| | ImageNet-C | ResNet50 | 38.38 | 0.1 | **38.70** | **38.64** | **38.72** |
| | ImageNet-C | ResNet50 | 40.38 | 0.1 | **41.44** | **40.50** | **41.07** |
| | ImageNet-3DCC | ResNet50 | 43.62 | 0.1 | **44.08** | **43.87** | **44.46** |
| | ImageNet-C | DeiT | 54.68 | 0.1 | **54.73** | **55.15** | **54.94** |
| videos | UCF101 | 3D ResNet50 | 51.14 | 0.1 | **52.71** | **53.00** | **53.57** |
| | UCF101 | I3D | 53.58 | 0.1 | **55.00** | **56.3** | **55.21** |
| Total wins | | | 0 | | | 9 | |

the dynamic sparsity process performs a form of implicit regularization which makes the model to automatically learn to focus on the more informative features (e.g., rich in low-frequency), while paying less attention to the less informative ones (e.g., rich in high-frequency).

## ACKNOWLEDGEMENTS

Qiao Xiao is supported by the research program 'MegaMind - Measuring, Gathering, Mining and Integrating Data for Self-management in the Edge of the Electricity System', (partly) financed by the Dutch Research Council (NWO) through the Perspectief program under number P19-25. Elena Mocanu is partly supported by the Modular Integrated Sustainable Datacenter (MISD) project funded by the Dutch Ministry of Economic Affairs and Climate under the European Important Projects of Common European Interest - Cloud Infrastructure and Services (IPCEI-CIS) program for 2024-2029. This research used the Dutch national e-infrastructure with the support of the SURF Cooperative, using grant no. EINF-7499.

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

# A APPENDIX

## A.1 DATASET INTRODUCTION

### A.1.1 CORRUPTION DATASET

Image corruptions refer to visible distortions in images that lead to data distribution shifts from that of the original data. They are common in practical applications when models are deployed in the real world. Typical corruptions include Gaussian noise, impulse noise, defocus blur, etc. In Figure 8, we display the common image corruptions on the CIFAR-10 dataset.

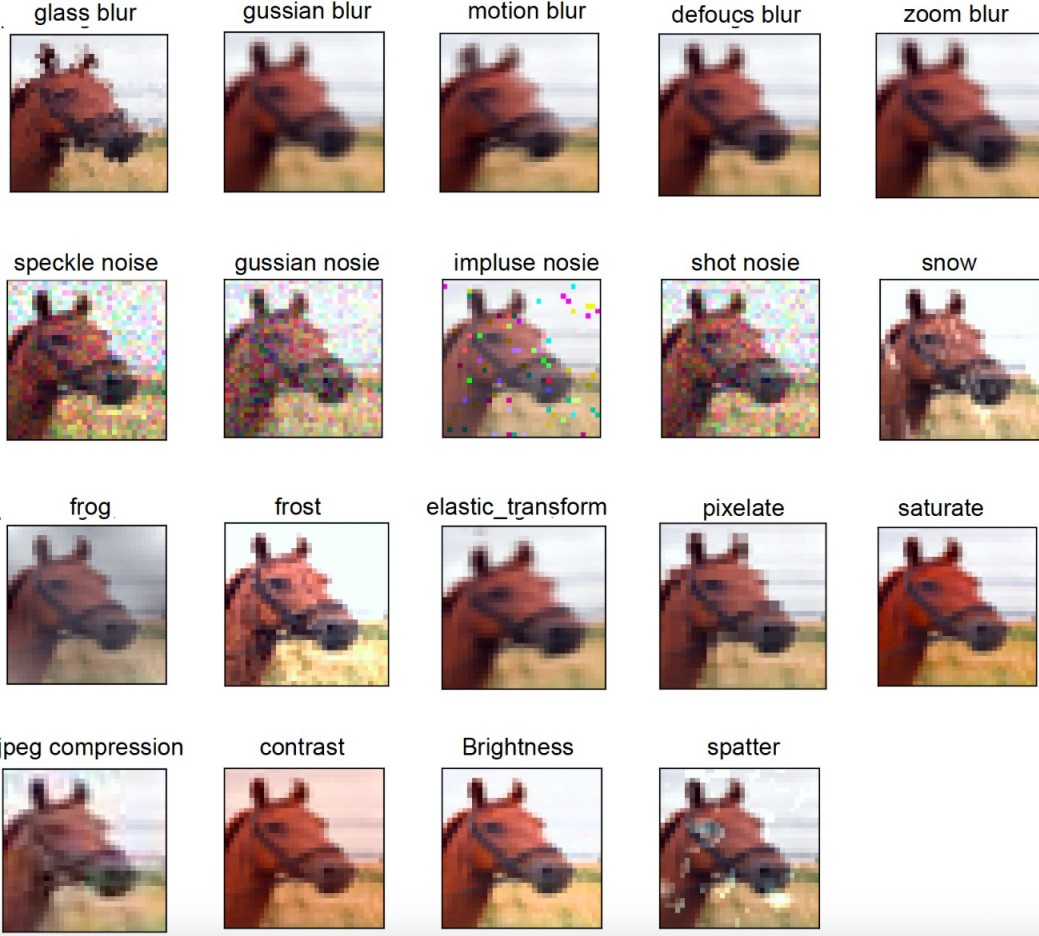

Figure 8: Visualization of the common image corruptions on CIFAR10 dataset.

## A.2 RELATED BENCHMARKS

- **CIFAR-10/100-C** Hendrycks & Dietterich (2019) is synthetically generated on top of the test set of CIFAR-10/100 dataset. It includes 19 sub-datasets, each corrupted with a type of image corruption (Gaussian noise, impulse noise, shot noise, speckle noise, defocus blur, Gaussian blur, glass blur, motion blur, zoom blur, brightness, fog, frost, snow, spatter, contrast, elastic transform, JPEG compression, pixelate, saturate). Each corruption dataset contains five subsets, which have images corrupted with five severity levels. The higher the severity, the more influence the corruption has on the test images.

- **Tiny-ImageNet-C** Hendrycks & Dietterich (2019) consists of 15 types of common image corruptions, and is synthetically generated from Tiny-ImageNet dataset. The types of corruption are Gaussian noise, impulse noise, shot noise, defocus blur, glass blur, motion

blur, zoom blur, brightness, fog, frost, snow, contrast, pixelate, elastic transform, and JPEG compression, with 5 severity levels.

- **ImageNet-C** Hendrycks & Dietterich (2019) consists of 15 types of common image corruptions, and is synthetically generated from ImageNet dataset. The types of corruption are Gaussian noise, impulse noise, shot noise, defocus blur, glass blur, motion blur, zoom blur, brightness, fog, frost, snow, contrast, pixelate, elastic transform, and JPEG compression, with 5 severity levels.

- **ImageNet-C̄** Mintun et al. (2021) are perceptually dissimilar to ImageNet-C in our transform feature space, consists of 10 types of common image corruptions. The types of corruption are blue noise, brownish noise, caustic refraction, checkerboard cutout, cocentric sine waves, inverse sparkles, perlin noise, plasma noise, single frequency greyscale, sparkles, with 5 severity levels.

- **ImageNet-3DCC** Kar et al. (2022): addresses several aspects of the real world, such as camera motion, weather, occlusions, depth of field, and lighting. It consists of 11 types of common image corruptions: near focus, far focus, fog 3d, flash, color quant, low light, xy motion blur, z motion blur, iso noise, bit error, h265 abr, with 5 severity levels.

- **Corrupted UCF101** Soomro et al. (2012): a realistic action recognition dataset featuring videos collected from YouTube, spanning 101 different action categories. It consists of 16 corruptions: motion blur, compression, defocus blur, gaussian noise, shot noise, impulse noise, zoom blur, rotate, static rotate, speckle noise, sampling, reverse sampling, jumble, box jumble, freeze, translate, with 4 severity levels.

## A.3 EXPERIMENT SETUP

### A.3.1 IMPLEMENTATION DETAILS

For CIFAR10, CIFAR100, and TinyImageNet, the codes were implemented in Python using PyTorch and executed on a single NVIDIA A100 GPU. Each was run three times using three fixed random seeds. The ImageNet experiments were run using distributed computing on four NVIDIA A100 GPUs. A batch size of 64 was utilized for each task, resulting in a combined total batch size of 256. For the UCF101 dataset, we followed the settings described in Schiappa et al. (2023), utilizing the codebase [5]. For the FashionMNIST dataset, we followed the training settings detailed in Zhang et al. (2024b;a), using the codebase [6].

Table 3: The experimental hyperparameters settings. The hyperparameters include Learning Rate (LR), Batch Size (BS), LR Schedule (LRS), Optimizer (Opt), Weight Decay (WD), Sparsity Distribution (Sparsity Dist), Topology Update Interval ($\Delta T$), Pruning Rate Decay Schedule (Sched), Initial Pruning Rate (P).

| Model | Data | # Epoch | LR | BS | LRS | Opt | WD | Sparsity Dist [2] | $\Delta T$ | Sched [3] | P |
|---|---|---|---|---|---|---|---|---|---|---|---|
| VGG16 | CIFAR10 | 160 | 0.1 | 100 | Cosine | SGD [4] | 5e-4 | Uniform | 500 | polynomial | 0.1 |
| ResNet34 | CIFAR100 | 160 | 0.1 | 100 | Cosine | SGD | 5e-4 | Uniform | 500 | polynomial | 0.1 |
| EfficientNet-B0 | TinyImageNet | 100 | 0.1 | 128 | Cosine | SGD | 5e-4 | Uniform | 100 | polynomial | 0.1 |
| ResNet50 | ImageNet | 90 | 0.1 | 256 | Step [1] | SGD | 1e-4 | ERK | 2000 | polynomial | 0.1 |
| DeiT-base | ImageNet | 200 | 0.0005 | 512 | Cosine | AdamW | 5e-2 | Uniform | 7000 | polynomial | 0.5 |
| 3D ResNet50/I3D | UCF101 | 80 | 0.1 | 8 | Cosine | SGD | 1e-3 | ERK | 300 | polynomial | 0.1 |
| MLP | Fashion-MNIST | 100 | 0.025 | 8 | Cosine | SGD | 5e-4 | ERK | 300 | polynomial | 0.1 |
| FCN-VGG16 | Pascal VOC 2012 | 80 | 0.0001 | 4 | polynomial | SGD | 1e-4 | ERK | 500 | cosine | 0.5 |

[1] Step denotes a schedule that the learning rate decayed by 10 at every 30 epochs until 90 epochs.
[2] For the uniform distribution, the sparsity ratio of each layer is the same (i.e. $S_i = 1 - \|\boldsymbol{\theta}_i'\|_0 / \|\boldsymbol{\theta}_i\|_0$), where $\boldsymbol{\theta}_i$ is the parameters in $i$-th layer. ERK distribution allocates higher sparsities to layers with more parameters, while assigning lower sparsities to smaller layers. In ERK, the number of parameters of the sparse convolutional layers is scaled proportionally to $1 - \frac{n_{i-1}+n_i+w_i+h_i}{n_{i-1} \times n_i \times w_i \times h_i}$, where $n_{i-1}$ and $n_i$ are the number of input channels and output channels in $i$-th layer, $w_i$ and $h_i$ are the width and the height of the kernel.
[3] The remove and regrow ratio is decayed according to a polynomial strategy: $p \times \left(1 - \frac{\text{step}_{\text{current}}}{\text{step}_{\text{total}}}\right)^{0.01}$, where $\text{step}_{\text{current}}$ is the current step and $\text{step}_{\text{total}}$ is the total number of training steps.
[4] SGD optimizer with momentum 0.9.

[5] https://github.com/Maddy12/ActionRecognitionRobustnessEval
[6] https://github.com/biomedical-cybernetics/Cannistraci-Hebb-training

A.3.2 THE COMPUTATIONAL AND MEMORY CONSUMPTION

In this section, we present the computational cost in terms of Floating-Point Operations (FLOPs) for training and inference, as well as the memory cost in terms of the number of parameters, for different models across different sparsity ratios.

**CIFAR-10:** For the CIFAR10 dataset, we train VGG16 for 160 epochs. The soft memory bound for MEST is set to 10% of the target density. For example, if the sparse network's density is 0.3, then the soft memory bound is 0.03, resulting in an initial model density of 0.33. This soft memory bound is decayed to 0 using a polynomial strategy. The initial density of GraNet is set at 0.8. Every 500 training steps, it prunes weights to achieve a density described by the formula: $d_i - (d_i - d_t) \left(1 - \frac{t - t_0}{n\Delta t}\right)^3$, where $d_i$ is the initial density, $d_t$ is the target density, and $n\Delta t$ is set to half of the total training time.

Table 4: Training FLOPs (TrFLOPs, $\times 10^{16}$), Inference FLOPs (InFLOPs, $\times 10^9$), and the number of model parameters (Param, $\times 10^6$) for VGG16 on CIFAR10 at various sparsity ratios ($S$) using different DST algorithms.

| | TrFLOPs ($\times 10^{16}$) | | | | | | InFLOPs ($\times 10^8$) | | | | | | Param ($\times 10^6$) | | | | | |
| --- | --- | --- | --- | --- | --- | --- | --- | --- | --- | --- | --- | --- | --- | --- | --- | --- | --- | --- |
| $S$ | 0.7 | 0.6 | 0.5 | 0.4 | 0.3 | 0.0 | 0.7 | 0.6 | 0.5 | 0.4 | 0.3 | 0.0 | 0.7 | 0.6 | 0.5 | 0.4 | 0.3 | 0.0 |
| Dense | - | - | - | - | - | 1.51 | - | - | - | - | - | 6.30 | - | - | - | - | - | 15.25 |
| SET / RigL | 0.46 | 0.61 | 0.76 | 0.91 | 1.06 | - | 1.92 | 2.55 | 3.17 | 3.80 | 4.42 | - | 4.95 | 6.43 | 7.89 | 9.37 | 10.81 | - |
| MEST$_r$ / MEST$_g$ | 0.51 | 0.67 | 0.84 | 1.00 | 1.17 | - | 1.92 | 2.55 | 3.17 | 3.80 | 4.42 | - | 4.95 | 6.43 | 7.89 | 9.37 | 10.81 | - |
| GraNet$_r$ / GraNet$_g$ | 0.68 | 0.73 | 0.85 | 0.97 | 1.09 | - | 1.92 | 2.55 | 3.17 | 3.80 | 4.42 | - | 4.95 | 6.43 | 7.89 | 9.37 | 10.81 | - |

**CIFAR-100:** For the CIFAR100 dataset, we train ResNet34 for 160 epochs. The soft memory bound for MEST is set to 10% of the target density. For example, if the sparse network's density is 0.3, then the soft memory bound is 0.03, resulting in an initial model density of 0.33. This soft memory bound is decayed to 0 using a polynomial strategy. The initial density of GraNet is set at 0.8. Every 500 training steps, it prunes weights to achieve a density described by the formula: $d_i - (d_i - d_t) \left(1 - \frac{t - t_0}{n\Delta t}\right)^3$, where $d_i$ is the initial density, $d_t$ is the target density, and $n\Delta t$ is set to half of the total training time.

Table 5: Training FLOPs (TrFLOPs, $\times 10^{16}$), Inference FLOPs (InFLOPs, $\times 10^9$), and the number of model parameters (Param, $\times 10^6$) for ResNet34 on CIFAR100 at various sparsity ratios ($S$) using different DST algorithms.

| | TrFLOPs ($\times 10^{16}$) | | | | | | InFLOPs ($\times 10^9$) | | | | | | Param ($\times 10^6$) | | | | | |
| --- | --- | --- | --- | --- | --- | --- | --- | --- | --- | --- | --- | --- | --- | --- | --- | --- | --- | --- |
| $S$ | 0.7 | 0.6 | 0.5 | 0.4 | 0.3 | 0.0 | 0.7 | 0.6 | 0.5 | 0.4 | 0.3 | 0.0 | 0.7 | 0.6 | 0.5 | 0.4 | 0.3 | 0.0 |
| dense | - | - | - | - | - | 5.57 | - | - | - | - | - | 2.32 | - | - | - | - | - | 21.33 |
| SET / RigL | 1.68 | 2.24 | 2.79 | 3.35 | 3.90 | - | 0.67 | 0.93 | 1.18 | 1.40 | 1.62 | - | 6.45 | 8.57 | 10.70 | 12.82 | 14.95 | - |
| MEST$_r$ / MEST$_g$ | 1.85 | 2.46 | 3.07 | 3.68 | 4.29 | - | 0.67 | 0.93 | 1.18 | 1.40 | 1.62 | - | 6.45 | 8.57 | 10.70 | 12.82 | 14.95 | - |
| GraNet$_r$ / GraNet$_g$ | 2.51 | 2.68 | 3.13 | 3.63 | 4.02 | - | 0.67 | 0.93 | 1.18 | 1.40 | 1.62 | - | 6.45 | 8.57 | 10.70 | 12.82 | 14.95 | - |

**TinyImageNet:** For the TinyImageNet dataset, we train EfficientNet-B0 for 100 epochs. The soft memory bound for MEST is set to 10% of the target density. For example, if the sparse network's density is 0.3, then the soft memory bound is 0.03, resulting in an initial model density of 0.33. This soft memory bound is decayed to 0 using a polynomial strategy. The initial density of GraNet is set at 0.8. Every 500 training steps, it prunes weights to achieve a density described by the formula: $d_i - (d_i - d_t) \left(1 - \frac{t - t_0}{n\Delta t}\right)^3$, where $d_i$ is the initial density, $d_t$ is the target density, and $n\Delta t$ is set to half of the total training time.

Table 6: Training FLOPs (TrFLOPs, $\times 10^{15}$), Inference FLOPs (InFLOPs, $\times 10^9$), and the number of model parameters (Param, $\times 10^6$) for EfficientNet-B0 on TinyImageNet at various sparsity ratios ($S$) using different DST algorithms.

| | TrFLOPs ($\times 10^{15}$) | | | | | | InFLOPs ($\times 10^9$) | | | | | | Param ($\times 10^6$) | | | | | |
| --- | --- | --- | --- | --- | --- | --- | --- | --- | --- | --- | --- | --- | --- | --- | --- | --- | --- | --- |
| $S$ | 0.7 | 0.6 | 0.5 | 0.4 | 0.3 | 0.0 | 0.7 | 0.6 | 0.5 | 0.4 | 0.3 | 0.0 | 0.7 | 0.6 | 0.5 | 0.4 | 0.3 | 0.0 |
| Dense | - | - | - | - | - | 1.98 | - | - | - | - | - | 6.63 | - | - | - | - | - | 4.26 |
| SET/ RigL | 0.66 | 0.85 | 1.04 | 1.23 | 1.42 | - | 2.21 | 2.85 | 3.48 | 4.12 | 4.75 | - | 1.89 | 2.29 | 2.68 | 2.68 | 3.08 | - |
| MEST$_r$ / MEST$_g$ | 0.72 | 0.92 | 1.13 | 1.34 | 1.55 | - | 2.21 | 2.85 | 3.48 | 4.12 | 4.75 | - | 1.89 | 2.29 | 2.68 | 2.68 | 3.08 | - |
| GraNet$_r$ / GraNet$_g$ | 0.89 | 1.02 | 1.15 | 1.32 | 1.46 | - | 2.21 | 2.85 | 3.48 | 4.12 | 4.75 | - | 1.89 | 2.29 | 2.68 | 2.68 | 3.08 | - |

**ImageNet:** For the ImageNet dataset, we train ResNet50 for 90 epochs. The soft memory bound for MEST is set to 0.2, and the sparse network's density is 0.9, then the soft memory bound is 0.92, resulting in an initial model density of 0.92. This soft memory bound is decayed to 0 using a polynomial strategy. The initial density of GraNet is set at 0.95. Every 2000 training steps, it prunes weights to achieve a density described by the formula: $d_i - (d_i - d_t)\left(1 - \frac{t-t_0}{n\Delta t}\right)^3$, where $d_i$ is the initial density, $d_t$ is the target density, and $n\Delta t$ is set to half of the total training time.

Table 7: Training FLOPs (TrFLOPs, $\times 10^{18}$), Inference FLOPs (InFLOPs, $\times 10^9$), and the number of model parameters (Param, $\times 10^6$) for ResNet50 on ImageNet at sparsity ratios ($S = 0.1$) using different DST algorithms.

| | TrFLOPs ($\times 10^{18}$) | InFLOPs ($\times 10^9$) | Param ($\times 10^6$) | Init density $\rightarrow$ Final density |
|---|---|---|---|---|
| Dense | 2.28 | 8.21 | 25.56 | 1.0 |
| RigL | 2.17 | 7.80 | 23.00 | $0.9 \rightarrow 0.9$ |
| MEST$_g$ | 2.18 | 7.80 | 23.00 | $0.92 \rightarrow 0.9$ |
| GraNet$_g$ | 2.20 | 7.80 | 23.00 | $0.95 \rightarrow 0.9$ |

**The Resource Consumption and Robustness.** Even if it is outside of the scope of this paper, in the traditional DST style, we provide an overview of the relationship between resource efficiency and the model's robustness against common corruption. Figure 9 showcases the computational cost (i.e. training FLOPs and model sizes) and robustness of dense and dynamic sparse models on TinyImageNet-C (results for other datasets can be found in the above.

We can find that the dense model, with the highest resource usage (e.g., FLOPs and parameter count), does not necessarily lead to better robustness. In the end, we also investigate other training paradigms: iterative dense and sparse training, represented by AC/DC Peste et al. (2021) algorithm. AC/DC begins with a dense warm-up, then starts alternating dense and sparse training, returning accurate sparse-dense model pairs at the end of the training process. We find that AC/DC also exhibits decent robustness, as in Figure 9. However, in most cases, AC/DC tends to require more training FLOPs compared with DST methods.

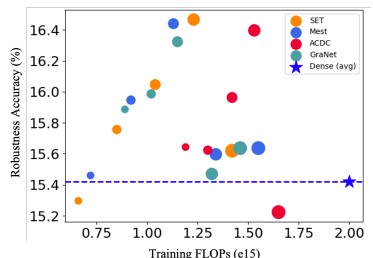

Figure 9: The comparison of training FLOPs, parameter count (represented by the area of each circle), and robustness accuracy of the dense and sparse EfficientNet-B0 on TinyImageNet. Each point represents the result for a specific sparsity ratio.

It is worth noting that the computational costs in this work are theoretical and not fully realized gains on current hardware infrastructures. This is a common limitation in the literature on sparse training methods, as most current hardware and software systems are optimized for dense matrix operations rather than sparse ones. However, recent advancements in model sparsity are increasingly aligning with hardware and software developments to fully leverage the benefits of sparsity. For example, NVIDIA's A100 GPU supports 2:4 sparsity Zhou et al. (2021), and other innovations are making strides toward efficient sparse implementations Chen et al. (2019); Ashby et al. (2019). Simultaneously, software libraries such as Liu et al. (2021b); Mocanu et al. (2018) are emerging to enable truly sparse network implementations. These advancements are paving the way for future deep neural networks to achieve greater efficiency in terms of computation, memory, and energy use.

## A.4 THE ROBUSTNESS ACCURACY FOR DIFFERENT CORRUPTION TYPES AND SERVERITIES

Figure 10, 11, 12, 13 14 and 15 showcase the relative gains[7] in robustness at each of these severity levels. We observe that dynamic sparse models exhibit a more pronounced advantage in robustness over dense models, particularly under high severity levels of high-frequency corruption.

---

[7]Defined as $(Acc_{s,l} - Acc_{dense})/Acc_{d,l}$, where $Acc_{s,l}$ and $Acc_{d,l}$ represent the accuracies of dynamic sparse and dense models, respectively, under severity level $l$.

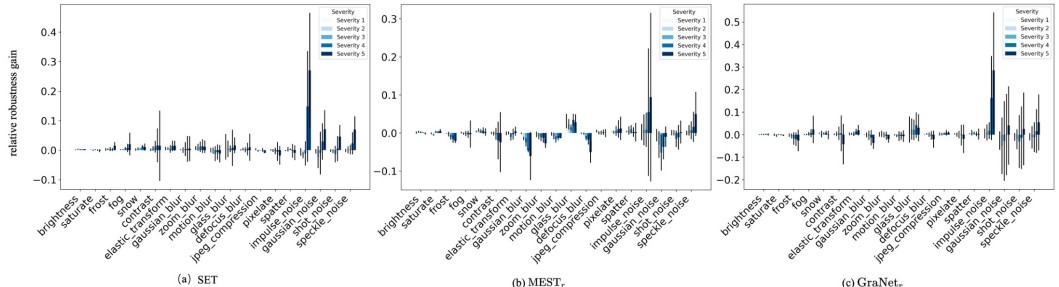

Figure 10: Compared with the dense model, the relative robustness gain of different dynamic sparse models with sparsity $0.5$ trained by (a) SET, (b) MEST$_r$, (c) GraNet$_r$ under different severities of CIFAR10-C.

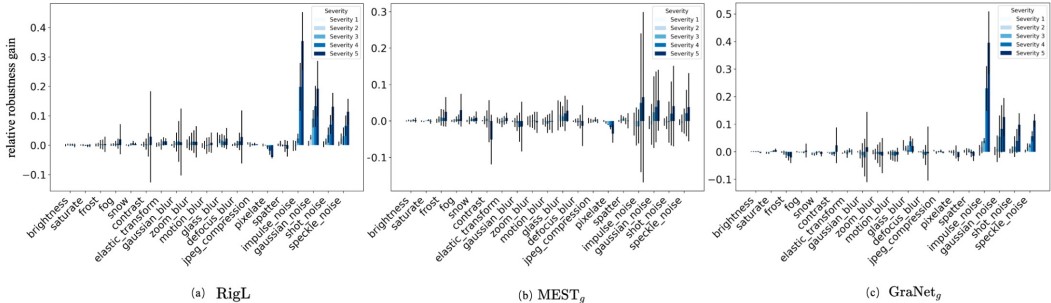

Figure 11: Compared with the dense model, the relative robustness gain of different dynamic sparse models with sparsity $0.5$ trained by (a) RigL, (b) MEST$_g$, (c) GraNet$_g$ under different severities of CIFAR10-C.

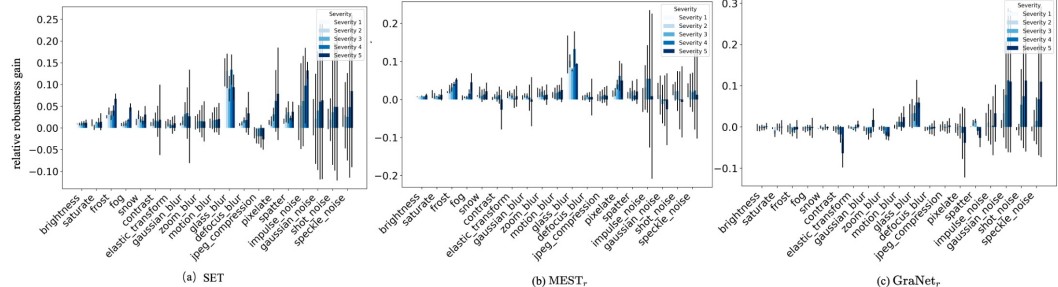

Figure 12: Compared with the dense model, the relative robustness gain of different dynamic sparse models with sparsity $0.5$ trained by (a) SET, (b) MEST$_r$, (c) GraNet$_r$ under different severities of CIFAR100-C.

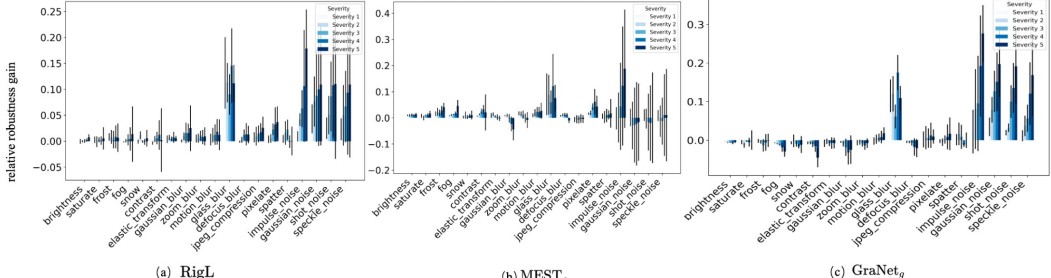

Figure 13: Compared with the dense model, the relative robustness gain of different dynamic sparse models with sparsity $0.5$ trained by (a) RigL, (b) MEST$_g$, (c) GraNet$_g$ under different severities of CIFAR100-C.

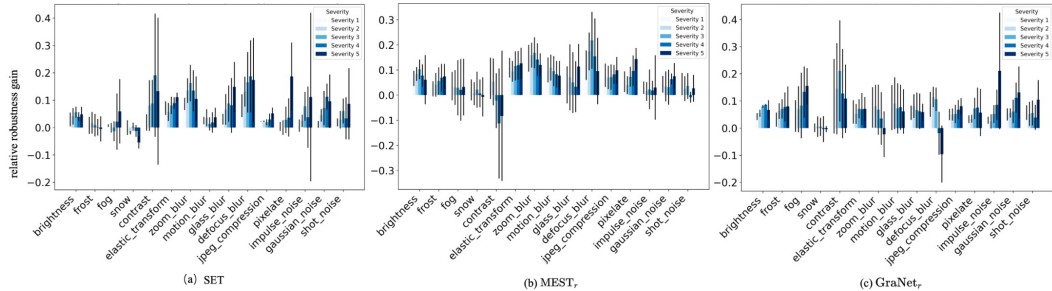

Figure 14: Compared with the dense model, the relative robustness gain of different dynamic sparse models with sparsity $0.5$ trained by (a) SET, (b) MEST$_r$, (c) GraNet$_r$ under different severities of TinyImageNet-C.

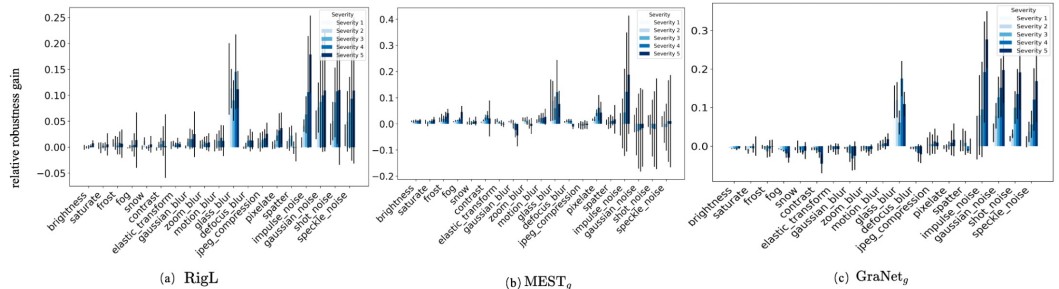

Figure 15: Compared with the dense model, the relative robustness gain of different dynamic sparse models with sparsity $0.5$ trained by (a) RigL, (b) MEST$_g$, (c) GraNet$_g$ under different severities of TinyImageNet-C.

## A.5 ANALYSIS THROUGH THE LENS OF FILTER

Figure 16 and 17 showcases the non-zero weight counts and the sum of weight magnitudes (i.e., absolute values) for each kernel in a specific layer of dynamic sparse VGG16 and ResNet34 trained on CIFAR10 and CIFAR100, respectively.

Additionally, we investigated the accuracy of test data subject to high-frequency or low-frequency information attenuation during dense or dynamic sparse training. In Figure 18, the results indicate that at the beginning of training, the superiority of dynamic sparse models in handling high-frequency information removal starts to appear, becoming more evident as training progresses. Considering the phenomenon of sparse weight concentration on the filter after training, it becomes evident that dynamic sparse training dynamically allocates computational resources to prioritize relevant low-frequency features during the learning process.

## A.6 DSCR HYPOTHESIS IN RECENT DST ALGORITHMS

In this paper, we primarily investigate the robustness of fundamental DST models initialized from random sparse networks with the basic removal and regrowth

Table 8: Robustness accuracy (%) of MLP on Fashion MNIST-C.

| Dense Training | Dynamic Sparse Training | | | | |
|---|---|---|---|---|---|
| | Sparsity | SET | RigL | CHTs+CSTI | CHTs+BSW |
| $19.82 \pm 0.32$ | 97% | $23.44 \pm 1.79$ | $25.23 \pm 1.39$ | $24.50 \pm 1.03$ | $\mathbf{26.64 \pm 1.06}$ |
| | 95% | $25.55 \pm 0.32$ | $25.03 \pm 1.11$ | $\mathbf{27.38 \pm 0.25}$ | $27.13 \pm 0.33$ |

criteria, to validate our DSCR hypothesis in commonly used settings. With the emergence of other recent and promising DST methods, we also include an initial exploration to further validate our hypothesis on these methods, such as Cannistraci-Hebb soft training (CHTs) Zhang et al. (2024b;a), which employ more complex growth criteria and topology initialization. We compare dense training with CHTs using Correlated Sparse Topological Initialization (CHTs+CSTI) and bipartite small-world networks (CHTs+BSW) on robustness accuracy, using the Fashion-MNIST corrupted dataset with an MLP model, as shown in Table 8. More training details are provided in Appendix A.3.1.

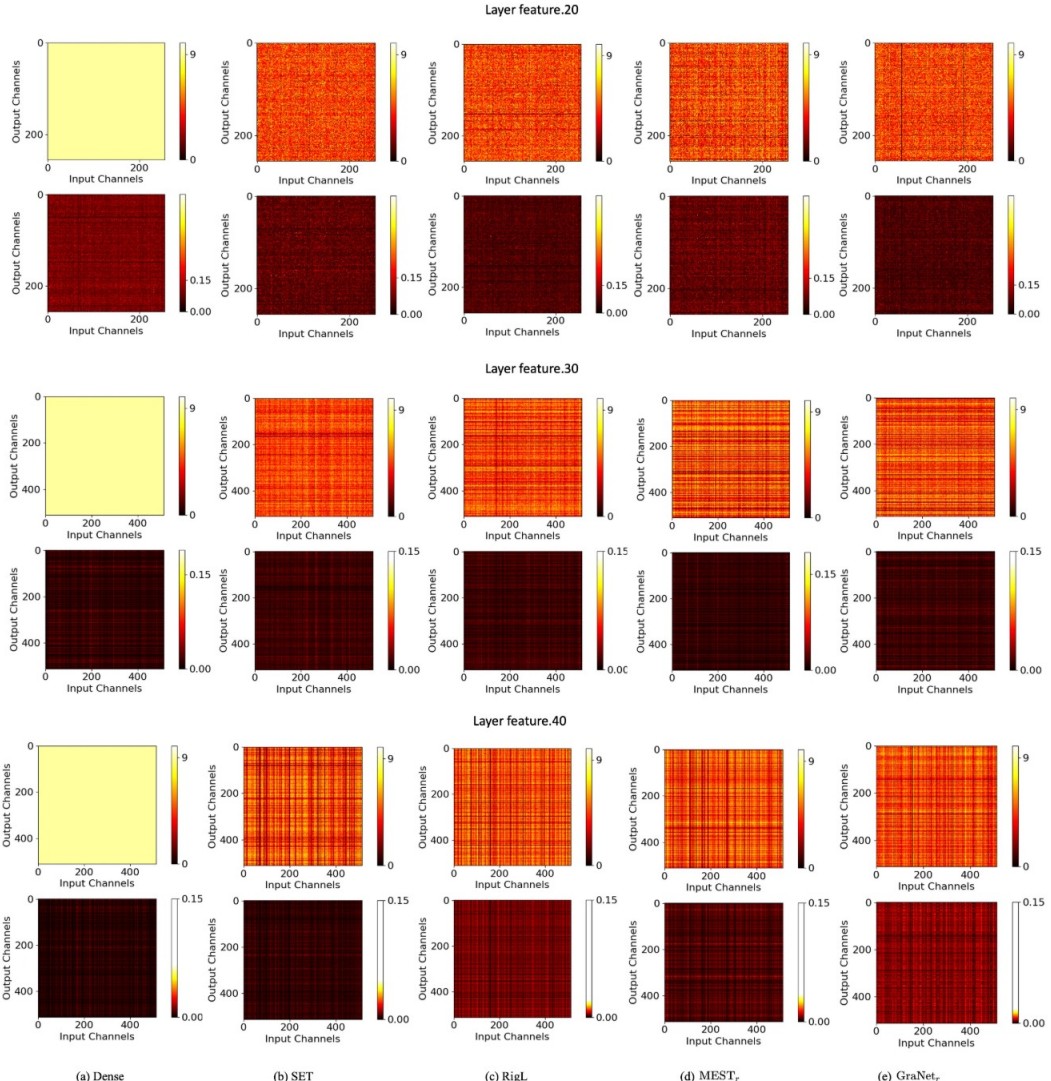

Figure 16: Visualizing non-zero weight count (1st row) and the sum of weight magnitudes (2nd row) in filters of VGG16 (#layer feature.20, #layer feature.30 and #layer feature.40 with kernel size $3 \times 3$) after training on CIFAR10 using different DST algorithms: (b) SET, (c) RigL, (d) MEST$_r$ and (e) GraNet$_r$, compared with (a) dense counterpart. Each point represents the corresponding value of a filter.

The results indicate that the recent DST method (i.e., CHTs) achieves even more inspiring robustness performance, further validating our DSCR hypothesis on the robustness of DST.

## A.7 HOW DATA AUGMENTATION INTERACTS WITH DST?

We extend our analysis to explore how data augmentation (e.g., Mixup Zhang et al. (2018)) interacts with dynamic sparse training in comparison to dense training. From Table 9, we observe: In general, data augmentation helps improve model generalization by providing increased input variability. For DST, combining data augmentation with dynamic sparse training can lead to even greater robustness compared to dense training, suggesting that dynamic sparse training (e.g., SET) interacts with data augmentation in a complementary way.

Table 9: Robustness accuracy (%) on CIFAR100 with and without Mixup.

| w/ mixup | Dense training | SET |
|---|---|---|
| $\times$ | 51.68 | **52.60** |
| $\checkmark$ | 54.75 | **54.93** |

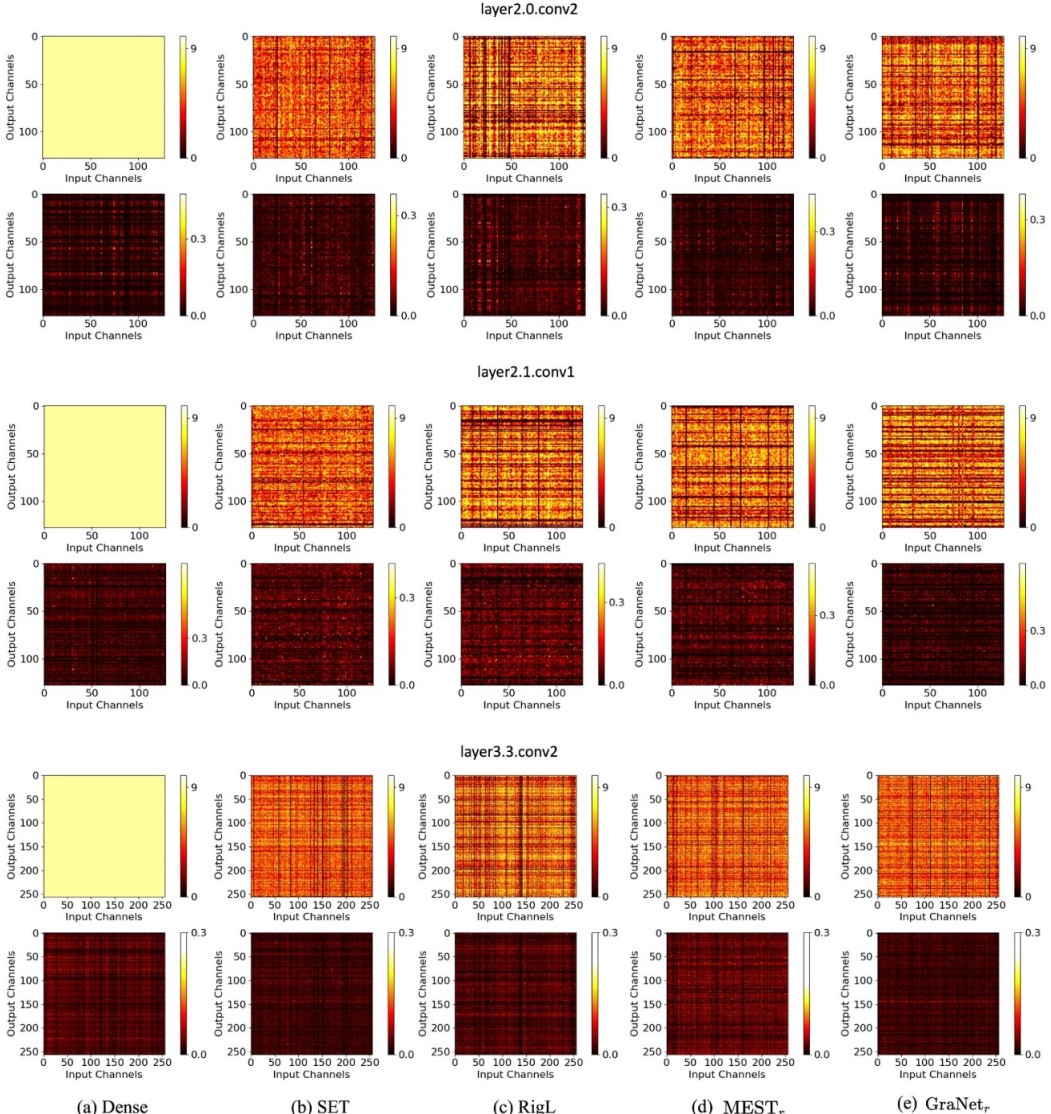

Figure 17: Visualizing non-zero weight count (1st row) and the sum of weight magnitudes (2nd row) in filters of ResNet34 (#layer2.0.conv2, #layer2.1.conv1 and #layerlayer3.3.conv2 with kernel size $3 \times 3$) after training on CIFAR100 using different DST algorithms: (b) SET, (c) RigL, (d) MEST$_r$ and (e) GraNet$_r$, compared with (a) dense counterpart. Each point represents the corresponding value of a filter.

## A.8 THE PERFORMANCE OF DST ON OUT-OF-DOMAIN (OOD) TEST DATA

**ImageNet-R** Hendrycks et al. with 30,000 images of art, cartoons, and other 14 renditions from 200 ImageNet classes, presents a notable domain shift from the original dataset. We report the average test accuracy on ImageNet-R for dense and DST models trained on the original ImageNet.

**ImageNet-v2** Recht et al. is a valuable dataset for evaluating the generalization of models trained on original ImageNet dataset. ImageNet-v2 contains three test subsets: Top-images, Threshold-0.7, and Matched-frequency. Each subset comprises ten images per class, selected from a pool of candidates according to various selection frequencies.

Table 10: Average classification accuracy (%) on ImageNet-R dataset. The bold highlights result that are better than the dense model.

| Dense | RigL | MEST$_g$ | GraNet$_g$ |
|---|---|---|---|
| 37.84 | **40.04** | **38.33** | **38.46** |

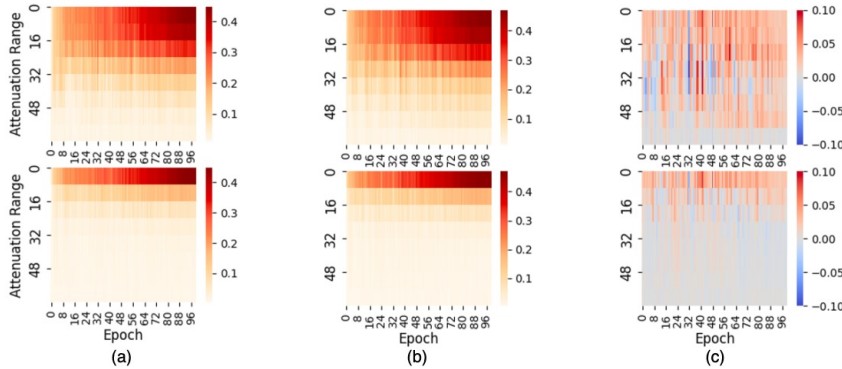

Figure 18: The impact of high-frequency (top) and low-frequency attenuation (bottom) on TinyImageNet during training. Test accuracy for (a) dense model, (b) dynamic sparse model from SET. (c) The accuracy difference between sparse and dense models, where a positive value indicates that the sparse models perform better.

As shown in Table 10 and 11, the results from ImageNet-R and ImageNet-v2 correspond to the evaluation on domain adaptation and generalization, respectively. The results show that DST models reliably surpass dense models on ImageNet-R and exhibit superior performance in the majority of scenarios on ImageNet-v2. This suggests that models trained with DST not only handle common data corruptions effectively but also excel in the face of significant domain shifts.

Table 11: Average robustness accuracy (%) on ImageNet-v2 dataset. The bold highlights result that are better than the dense model.

|  | Dense | RigL | $MEST_g$ | $GraNet_g$ |
|---|---|---|---|---|
| Top-images | 77.80 | **77.88** | **78.33** | **78.21** |
| Threshold-0.7 | 72.95 | 72.81 | **73.23** | **73.58** |
| Matched-frequency | 64.62 | 64.23 | 64.14 | 64.60 |
| Avg. accuracy | 71.79 | 71.64 | **71.90** | **72.13** |

## A.9 THE PERFORMANCE OF DST ON THE SEGMENTATION TASK

We compared the performance of a dense trained fully convolutional network (FCN) with a VGG16 backbone on the Pascal VOC 2012 Everingham et al. clean dataset, then test on corrupted Pascal VOC 2012 (generated following the ImageCorruption library [8], which includes 19 types of corruption at 5 severity levels). The results are presented in Table 12.

Unlike classification tasks on CIFAR or ImageNet, where the sparse topology is randomly initialized, we first prune the backbone to obtain the sparse model, then perform sparse fine-tuning on the training set of the Pascal VOC 2012 clean dataset, and finally test it on both the clean and corrupted versions of the Pascal VOC 2012 test set, as the VGG16 back-

Table 12: Clean metric (%) on Pascal VOC 2012 and Robustness metric (%) on corrupted Pascal VOC 2012.

| Metric | Dense training | SET (s=0.2) | SET (s=0.1) |
|---|---|---|---|
| clean pixAcc (↑) | 85.65 | 85.63 | 85.66 |
| clean mIoU (↑) | 46.34 | 46.69 | 47.10 |
| robustness pixAcc (↑) | 76.80 | 77.30 | 77.39 |
| robustness mIoU (↑) | 22.60 | 22.73 | 23.09 |

bone for the segmentor is typically pretrained on ImageNet. The results show that DST models are not only robust in image and video classification but also in image segmentation tasks.

## A.10 GRAD-CAM

Grad-CAM (Gradient-weighted Class Activation Mapping) Selvaraju et al. (2017) generates a heatmap highlighting important regions by weighting the feature maps based on the gradient values. It provides practitioners with a visual indication of the areas on which the network focuses when making predictions. We visualize and compare the Grad-CAM outputs of both dense models and dynamic sparse models, as shown in Figure 19. We present different cases, including clean background ((a) and (b)), and complex background (c). Interestingly, we observe that the highlighting region of dynamic sparse models is smaller than that of the dense model. In Figure 19 (a), the dynamic sparse models focus on the face of the cat for prediction, while the dense models pay attention to both the face and body of the cat. In Figure 19 (b), the dynamic sparse models specifically

---

[8]https://github.com/bethgelab/imagecorruptions

emphasize the wings of the eagle. In contrast, the dense models primarily concentrate on a single wing while also capturing irrelevant contexts, leading to larger attention regions. In the case of a more complex background (In Figure 19 (c)), the dynamic sparse models tend to allocate some attention to the surrounding elements. However, the attention regions are still relatively smaller and primarily focused on the face of the bear, especially in the sparse model sparsified by SET.

These visualizations provide further support for our hypothesis that dynamic sparse models effectively allocate limited resources to the most crucial features. As a result, dynamic sparse models can be robust against image common corruptions, making them a promising approach for addressing real-world challenges.

## B   IMPACT STATEMENT

In an era dominated by over-parameterized models and the pervasive presence of image corruption, the design of resource-aware and robust AI models is of increasing importance. Gaining insight into the robust behavior of dynamic sparse models against image corruption paves the way for embracing these environmentally friendly models in challenging environments. This holds significant implications across various domains, including medical diagnosis, robotics, autonomous vehicles, and other AI applications. Furthermore, our insights into the inner workings of sparse models help us understand the reasoning behind their robust decisions. Overall, we advance our fundamental understanding of dynamic sparse training and provide future perspectives for scalable, efficient, and trustworthy AI. We do not anticipate any negative societal impacts resulting from this research.

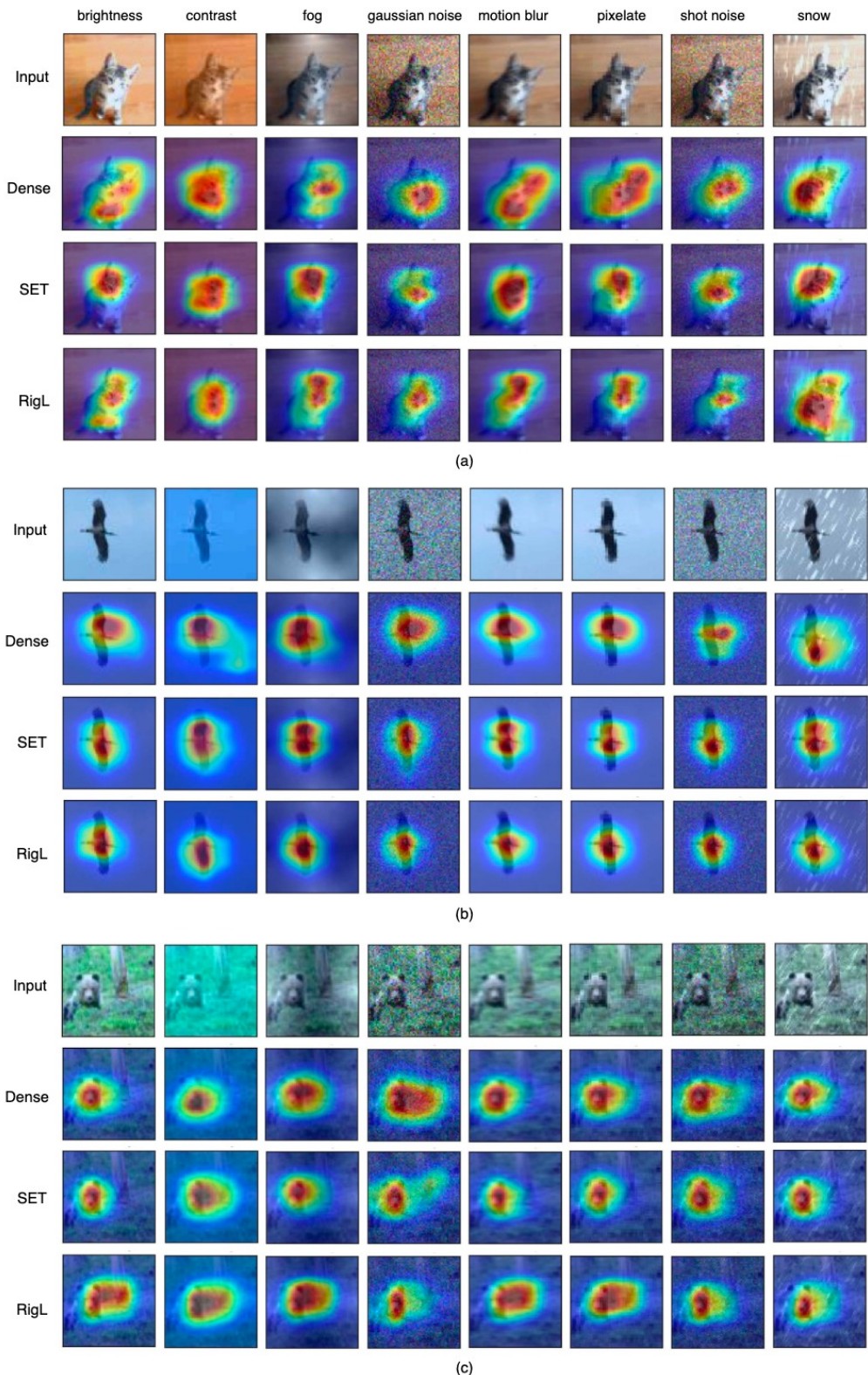

Figure 19: Grad-CAM for TinyImageNet-C. The images are corrupted by different corruptions with clean background (a) and (b), and complex background (c).

