# OpenReview forum: "Dynamic Sparse Training versus Dense Training: The Unexpected Winner in Image Corruption Robustness"
_ICLR.cc/2025/Conference — ICLR 2025 Poster_

### Official Review · Reviewer_Khz6 · 2024-11-01

**Soundness:** 2
**Presentation:** 2
**Contribution:** 2
**Rating:** 3
**Confidence:** 4

**Summary:**

This paper challenges the conventional use of dense training for model robustness in deep learning. Through extensive experiments across multiple datasets, architectures, and Dynamic Sparse Training (DST) methods, the authors demonstrate that DST can consistently outperform dense training in image corruption robustness. Analysis reveals that DST's superior performance stems from its implicit regularization mechanism, which naturally focuses on low-frequency features while reducing sensitivity to high-frequency components. The findings are validated across both image and video tasks, suggesting new directions for improving neural network robustness.

**Strengths:**

The experimental evaluation is thorough and well-structured, covering both image and video domains, multiple architectures (CNNs and Transformers), and various corruption types and severity levels.

**Weaknesses:**

1.	The core contribution appears to be an empirical observation, while the technical contributions are limited.
2.	While the results show improved robustness against corruption, the paper doesn't extensively analyze whether this comes at the cost of other aspects of model performance (e.g., clean accuracy).
3.	There is a clear error in describing Figure 3's axis. The text states "the Y-axis in Figure 3 represents different types of corruption" when it appears to be the X-axis that shows corruption types.
4.	The considered datasets to evaluate the proposed method are limited. There are a lot of robustness benchmarks, including ImageNet-A/R/P. It is unclear whether the proposed method is able to consistently improve robustness across diverse benchmarks.
5.	It would be interesting to compare with SOTA robust training methods, rather than only comparing with the standard dense training.
6.	The comparisons against SOTA robust training methods and robust models should be included, including RVT and FAN.
7.	The comparison of training cost should be reported.

**Questions:**

More experiments and comparisons with SOTA training methods or robust models would be beneficial.

---

> ### Author Response · Authors · 2024-11-24
> **To Reviewer Khz6**
>
> Thank you for your valuable and constructive review. We are glad that you think our paper is thorough and well-structured. Here we address your comments one by one.
>
> > **Comment1:** The core contribution appears to be an empirical observation, while the technical contributions are limited.
>
> Thanks for your interest and for pointing this out. While we did not provide a theoretical analysis of the benefits of DST, **we offer insights from spatial and spectral perspectives to explain its superior performance.** Our findings suggest that DST **performs a form of implicit regularization**, guiding the model to focus on more important features (e.g., those rich in low-frequency) while diminishing attention to less important ones (e.g., those rich in high-frequency). We provide a detailed analysis of this phenomenon in Section 5.
>
> We believe this work is quite comprehensive, but the theoretical understanding of this phenomenon should be explored in our future research. We are happy to discuss and provide some directions for proving this finding. The theorem we aim to prove can be summarized as follows: **the DST network is less likely to learn high-frequency components.**
>
> 1. **Lemma 1**: We present the convolution operation in the frequency domain as: $(O_{u,v} = T_{u, v} \times I_{u, v})$. Then the magnitude of $(T_{u, v})$ can be used as an indicator to measure the strength of the network encoding the specific frequency component $((u,v))$.
> 2. **Theorem 1**: We provide a closed-form expression of the strength ratio of high and low frequency components: $(\frac{{T_{u, v}}^2}{{T_{u', v'}}^2})$, where $(u)$ and $(v)$ represent the high-frequency indices, and $(u')$ and $(v')$ represent the low-frequency indices. We find that the strength ratio is related to the kernel mean and variance.
> 3. **Lemma 2**: We compare the kernel mean and variance for dense training models and DST models.
>
> > **Comment2:** While the results show improved robustness against corruption, the paper doesn't extensively analyze whether this comes at the cost of other aspects of model performance (e.g., clean accuracy).
>
> Thank you for the constructive suggestion. We agree that providing the clean accuracy could give more insights. We present the results for ImageNet and ImageNet-C below. We found that on **in-domain data (corresponding to clean accuracy), dense models and DST models perform similarly. However, DST models demonstrate significantly improved robustness on out-of-domain (corrupted) data (corresponding to robustness accuracy)**. This further validates the effectiveness of DST models in enhancing robustness.
>
> Table. Clean accuracy (%) on ImageNet and  Robustness accuracy (%) on ImageNet-C.
>
> | Method   |  Model   | Clean acc | Robustness acc |
> | -------- | --- | --------- | -------------- |
> | Dense    |  DeiT-base   |    78.36    |       54.68         |
> | RigL     |  DeiT-base   |    78.30     |       **54.73**         |
> | MEST_g   |  DeiT-base   |    78.45     |       **55.15**        |
> | GraNet_g |  DeiT-base   |    78.36     |       **54.94**       |
>
>
> > **Comment3:** There is a clear error in describing Figure 3's axis. The text states "the Y-axis in Figure 3 represents different types of corruption" when it appears to be the X-axis that shows corruption types.
>
> Thank you for pointing this out. We will correct the description in the final version to accurately state that the X-axis in Figure 3 represents the different types of corruption.

---

> ### Author Response · Authors · 2024-11-24
> **To Reviewer Khz6**
>
> > **Comment4:** ....There are a lot of robustness benchmarks, including ImageNet-A/R/P. It is unclear whether the proposed method is able to consistently improve robustness across diverse benchmarks.
>
> We appreciate your valuable comment regarding the evaluation datasets. We have conducted extensive experiments to verify the DSCR Hypothesis across a wide variety of DST methods, models, and datasets. For datsets, these include not only image corruption benchmarks such as CIFAR10-C, CIFAR100-C, TinyImageNet-C, ImageNet-C, and ImageNet-$\bar{\mathbf{C}}$, but also video corruption benchmarks like corrupted-UCF101 under various severity levels.
>
> Regarding the other ImageNet benchmarks ImageNet-A/R/P, in our submission, we have included results for ImageNet-R and ImageNet-v2 **in Appendix A.7 (Table 9 and Table 11, respectively)**. These datasets evaluate domain adaptation and generalization capabilities. The results demonstrate that DST models reliably outperform dense models on ImageNet-R and achieve superior performance in most scenarios on ImageNet-v2. This highlights that DST models not only handle common data corruptions effectively but also excel under significant domain shifts.
>
> Additonally, we have also included results on the ImageNet-A dataset below. Due to time constraints during the rebuttal phase, results on ImageNet-P are not yet included. However, we are committed to updating the results during the discussion period if necessary.
>
> Table. Average classification accuracy (%) on ImageNet-A dataset. The bold highlights results that are better than the dense model.
> | Dense |RigL| MEST_g | GraNet_g |
> | - | - | - | - |
> | 3.15  | **3.27** | **3.24**| **3.29** |
> > **Comment5:** It would be interesting to compare with SOTA robust training methods, rather than only comparing with the standard dense training.
>
> Thank you for suggestions. We have conducted additional experiments using SET with a sparsity level of 0.8 and adversarial training as a robust training baseline. The results are summarized in the table below.
>
> These results suggest that sparse training techniques like SET, when combined with adversarial training, can effectively enhance both clean and adversarial performance. The comparison highlights the potential of DST methods to achieve competitive robustness while maintaining high accuracy.
>
> Table. Comparing clean and adversarial accuracy for both dense and sparse models on CIFAR10 and CIFAR100 datasets.
> |  |  CIFAR10(dense)| CIFAR10(sparse) | CIFAR100(dense) | CIFAR100(sparse) |
> | - | - | -| - | - |
> | clean_acc | 71.25 | **73.72** | 54.37| **55.25** |
> | adv_acc| 41.55 | **43.07**| 21.47| **23.46** |
>
> > **Comment6:** The comparisons against SOTA robust training methods and robust models should be included, including RVT and FAN.
>
> Thank you for your suggestion. Our study primarily challenges the conventional approach of using Dense Training to maximize model robustness and proposes Dynamic Sparse Training as a more effective alternative. It would be interesting to extend our analysis to **explore how SOTA models interact with Dynamic Sparse Training compared to Dense Training**.
>
> We dynamically trained the Robust Vision Transformer (RVT) [1] and Fully Attention Network (FAN) [2] with a sparsity level of $0.1$ using the RigL algorithm on the training set of ImageNet for 300 epochs, and tested them on the clean ImageNet test set and ImageNet-C. These results suggest that DST SOTA models, have the potencial to effectively enhance both clean and robustness performance.
>
> [1] Xiaofeng Mao, Gege Qi, Yuefeng Chen, Xiaodan Li, Ranjie Duan, Shaokai Ye, Yuan He, Hui Xue: Towards Robust Vision Transformer. CVPR 2022.
>
> [2] Understanding The Robustness in Vision Transformers.Daquan Zhou, Zhiding Yu, Enze Xie, Chaowei Xiao, Anima Anandkumar, Jiashi Feng and Jose M. Alvarez. ICML, 2022.
>
> Table. Clean accuracy (%) on ImageNet and  robustness accuracy (%) on ImageNet-C.
> |  |  RVT-Tiny  (dense) | RVT-Tiny (DST) | FAN-Tiny-Hybrid (dense) | FAN-Tiny-Hybrid (DST)|
> | - | -- | - | - | - |
> | clean acc |  78.5| 78.1| 79.9 | 80.2  |
> | robustness acc | 41.6 | **42.0**| 57.3| **57.6** |

---

> ### Author Response · Authors · 2024-11-24
> **To Reviewer Khz6**
>
> > **Comment7:** The comparison of training cost should be reported.
>
> Thank you for highlighting this point. In our submission, we have considered training costs in our analysis. Due to page limitations, we have included these details **in Appendix A.3.2**. Specifically, we report the computational cost in terms of Floating-Point Operations (FLOPs) for both training and inference, as well as the memory cost in terms of the number of parameters for different models across various sparsity ratios.
>
> Our findings show that models trained with sparse training not only reduce computational and memory costs but also demonstrate enhanced robustness against image corruption compared to dense training. We appreciate your suggestion and will ensure this is emphasized appropriately in the final version of the paper.
>
> Thank you for your feedback on our response. Please let us know if you still have any additional questions or would like further clarification. If you feel that we have addressed all of your concerns, we hope you might reconsider your score. Thanks again for your time and effort.

---

> > ### Comment · Reviewer_Khz6 · 2024-11-28
> > **Thanks for the response**
> >
> > After reading the response, my major concerns are still not well addressed.
> >
> > First, from the table provided for my comment2, the clean accuracy of dense training based on DeiT-Base should be 81.8 according to their paper, rather than 78.36. Since there is a large accuracy gap, it is unclear whether the authors trained the model with the correct setting. If the authors use a different setting (e.g. using different epochs or learning rate strategies), it is hard to believe whether this method still works well on a commonly used setting (with 300 epochs to train Deit).
> >
> > Second, the superiority over existing SOTA robust methods seems to disappear. From the table provided for my comment6, the accuracy of the dense training baseline is lower than the reported number of FAN-Tiny-Hybrid. For example, the accuracy should be 80.1 according to the original paper but the accuracy reported by the authors is 79.9. The same issue occurs once again. It would be better to clarify this. Moreover, for RVT-Tiny, DST incurs a large drop of 0.4 in terms of clean accuracy. From this point, DST seems not a strong method. It would help if the authors could further discuss this.
> >
> >
> > Based on the above, due to the relatively weak empirical results, we tend to lower my rating.
> >
> > Best,
> >
> > Reviewer Khz6

---

> ### Author Response · Authors · 2024-11-28
>
> Dear Reviewer Khz6,
>
> Thank you for your reply. We will address your concerns point by point.
>
> > First, from the table provided for my comment2, the clean accuracy of dense training based on DeiT-Base should be 81.8 according to their paper, rather than 78.36.  If the authors use a different setting (e.g. using different epochs or learning rate strategies), it is hard to believe whether this method still works well on a commonly used setting (with 300 epochs to train Deit).
>
> In our experiments, due to time constraints in the original deadline, we trained the DeiT-Base model on ImageNet2012 for 200 epochs, whereas the original paper trained it for 300 epochs. This difference is the primary reason for the observed performance discrepancy. **We have clarified this point in our main paper on line 346**.
>
> If you are concerned about the effectiveness of our method under commonly used settings (e.g., training DeiT-Base for 300 epochs), we will address this concern by conducting a comparison between DST and dense training under the same 300-epoch setting. We will share the updated results, which require several days of training, as soon as possible in the coming days.
>
> >  For example, the accuracy should be 80.1 according to the original paper but the accuracy reported by the authors is 79.9.
>
> Thank you for pointing this out. We believe there may be some variance due to differences in GPUs and seeds. Most importantly,  the **original RVT paper used 8 GPUs, while our experiments were conducted on 4 GPUs during rebuttal, due to limited device availability**. This difference in GPU settings could lead to minor discrepancies in performance.
>
> > The superiority over existing SOTA robust methods seems to disappear. Moreover, for RVT-Tiny, DST incurs a large drop of 0.4 in terms of clean accuracy. From this point, DST seems not a strong method. It would help if the authors could further discuss this.
>
> We respectfully disagree with the statement that 'the superiority over existing SOTA robust methods seems to disappear.' From Table 6, we observe a clear **improvement in robustness accuracy** when comparing DST to dense training methods for both **RVT-Tiny and FAN-Tiny-Hybrid**. While there is indeed a drop in clean accuracy for RVT-Tiny when using DST, our work highlights another critical, often overlooked perspective: the robustness of DST models, in contrast to prior DST studies that primarily focus on achieving comparable clean accuracy and improved efficiency.
>
> On the other hand, the lower clean accuracy alongside improved robustness demonstrates that **the enhanced robustness of DST is not merely a byproduct of improved clean accuracy** but rather stems from DST's inherent ability to 'pay less attention to high-frequency information compared to dense training.' This directly supports the main claim regarding the robustness of DST models. Furthermore, as observed in the table for DeiT referenced in Comment 2, the robustness accuracy gap between DST and dense training is larger than the clean accuracy gap, further reinforcing our claim.
>
> Best regards,
> Authors

---

> ### Author Response · Authors · 2024-12-02
>
> Dear Reviewer Khz6,
>
> Thank you for your comment. We have carefully considered your concerns and retrained DeiT-Base for 300 epochs to address the concerns.
>
> > First, from the table provided for my comment2, the clean accuracy of dense training based on DeiT-Base should be 81.8 according to their paper, rather than 78.36. Since there is a large accuracy gap, it is unclear whether the authors trained the model with the correct setting. If the authors use a different setting (e.g. using different epochs or learning rate strategies), it is hard to believe whether this method still works well on a commonly used setting (with 300 epochs to train Deit).
>
> Unfortunately, due to limited computational resources, we only have access to 4 GPUs for per task instead of the 8 GPUs used in the original DeiT paper. In the original setup, a total batch size of 1024 (8 GPUs). However, in our case, we were limited to a batch size of 512 (4 GPUs). Despite this limitation, we kept all other settings consistent with the default configurations provided in the original paper. However, our result on ImageNet2012 is 78.82 after training for 300 epochs, rather than 81.8.
>
> It is worth noting that several researchers have reported similar issues when reproducing DeiT-Base results. For example, a discussion on the DeiT GitHub repository (Issue #63: https://github.com/facebookresearch/deit/issues/63) highlights that the reproduced accuracy was 78.8% when the batch size was 512, compared to 81.8% with a batch size of 1024. Additionally, in Table 4 of another study [1], the authors reproduced DeiT-Base results as 80.9% (not 81.8%) under a similar setting of 8 GPUs (1 node) with a total batch size of 1024. Despite following nearly identical training configurations, discrepancies in results could also arise from differences in device setups (e.g., 1 node with 8 GPUs vs. 2 nodes with 4 GPUs in the original paper) and different pytorch versions. Such hardware-related variations and environments would cause fluctuations in performance.
>
> Although adjusting the learning rate scaling strategy, (i.e., adjusting learning rates based on the batch size [2]) or fine-tuning other hyperparameters might improve our reproduced results, we avoided such tuning to adhere to the original hyperparameters within the constraints of our available GPUs. This approach is particularly significant for researchers with constrained computational resources, as it offers a practical baseline for evaluation. Furthermore, **we maintained consistent hyperparameters for both dense and DST training to ensure a fair and unbiased comparison under the same limited-resource setup (4 GPUs, 512 batch size)**.
>
> Due to the limited time and resources, we rerun RigL and GraNet_g and dense training for 300 epochs, which cost ~3 days. The results are shown below. These results demonstrate that even with 300 epochs of training, DST models consistently outperform dense models, particularly in terms of robustness performance.
>
> Table. Clean accuracy (%) on ImageNet and  Robustness accuracy (%) on ImageNet-C.
>
> | Method   |  Model   | Clean acc | Robustness acc |
> | -| -| -| -|
> | Dense|  DeiT-base   |   78.82  |  55.51       |
> | RigL |  DeiT-base   | 78.94  |   **56.35** |
> | GraNet_g |  DeiT-base   |   78.85  |    **55.76**        |
>
> To further address your concern, 'While the results show improved robustness against corruption, the paper doesn't extensively analyze whether this comes at the cost of other aspects of model performance (e.g., clean accuracy),' as mentioned in Comment 2, we have additionally reported results for ResNet50 on ImageNet. The findings for both ResNet50 and DeiT-base demonstrate that **for in-domain data (corresponding to clean accuracy), DST models perform comparably to dense models. However, DST models consistently exhibit improved robustness on out-of-domain (corrupted) data (corresponding to robustness accuracy)**. These results further confirm that the improved robustness does not come at the expense of the model's clean performance.
>
> Table. Clean accuracy (%) on ImageNet and  Robustness accuracy (%) on ImageNet-C.
>
> | Method   |  Model   | Clean acc | Robustness acc |
> | -| -| -| -|
> | Dense|  ResNet50 |  76.36 |   38.38   |
> | RigL |  ResNet50 |   76.40 |  **38.70**  |
> | MEST_g|  ResNet50   |   76.41 |   **38.64**  |
> | GraNet_g |  ResNet50   |   76.45 |     **38.72** |
>
> *It is worth noting that our ResNet50 model was trained from scratch without using a pretrained model.*
>
> [1] Tianlong Chen, Yu Cheng, Zhe Gan, Lu Yuan, Lei Zhang, Zhangyang Wang. Chasing Sparsity in Vision Transformers: An End-to-End Exploration. NeurIPS 2021.
>
> [2] P Goyal, P Dollár, R Girshick, P Noordhuis, L Wesolowski, A Kyrola, A Tulloch, Y Jia, K He. Accurate, large minibatch sgd: Training imagenet in 1 hour. arXiv preprint arXiv:1706.02677, 2017.
>
> We kindly want to check if you feel that your concerns have been resolved, and we would greatly appreciate it if you could reconsider your score.

---

> ### Author Response · Authors · 2024-12-02
>
> Dear Reviewer Khz6,
>
> As the discussion period comes to a close, we would like to confirm that our responses have addressed your questions and concerns. We are confident that we have effectively addressed all key points and finer details in our replies. If there are any further questions or clarifications needed, we are eager to continue the discussion. If you find our responses satisfactory, we kindly request you to consider raising the score.
>
> Best regards,
>
> Authors

---

### Official Review · Reviewer_ncJc · 2024-11-02

**Soundness:** 3
**Presentation:** 3
**Contribution:** 3
**Rating:** 8
**Confidence:** 4

**Summary:**

This paper raises an intuitive research question through a thorough review of dense training, static sparse training, dynamic sparse training, and their counterpart model robustness, "Is somehow DST implicitly improving model robustness". Then, propose the Dynamic Sparsity Corruption Robustness (DSCR) Hypothesis "Dynamic Sparse Training at low sparsity levels helps to improve model robustness against image corruption in comparison with Dense Training." Finally, it validates the DSCR Hypothesis on two types of data, images, and videos, using several traditional and modern deep learning architectures for computer vision and three widely studied Dynamic Sparse Training algorithms. The findings reveal a new yet-unknown benefit of Dynamic Sparse Training and open new possibilities in improving deep learning robustness beyond the current state of the art. I think this is a good paper and worth accepting.

**Strengths:**

1. Problem statement. This paper clearly narrates the robustness from dense training, static sparse training to dynamic sparse training.
2. Necessary and sufficient experiments. From convolution-based network to transformer-based network, this paper makes a thoroughly empirical experiment to justify the robustness of dynamic sparse training.
3. Besides the above, section 5 explores the inner reasons for the robustness of dynamic sparse training models. Analyzing how DST and Dense training models utilize their parameters in the spatial domain and process the input in the spectral domain.

**Weaknesses:**

The research direction of robustness only focus on the classification task (with dataset cifar10/100-c, ImageNet-c), why not extend to new tasks/datasets like segmentation, pose estimation, and image matching et al?

**Questions:**

same as the weakness part.

---

> ### Author Response · Authors · 2024-11-24
> **To Reviewer ncJc**
>
> We sincerely thank you for your time and effort in reviewing our paper. We are pleased to hear that you appreciated the thorough experiments, the investigation into the underlying reasons behind robustness, and the valuable insights provided by our analysis. We address your comments below."
>
> > **Comment1:** The research direction of robustness only focus on the classification task (with dataset cifar10/100-c, ImageNet-c), why not extend to new tasks/datasets like segmentation, pose estimation, and image matching et al?
>
> Thank you for your suggestion. We agree that extending the research beyond classification tasks could provide broader insights into the robustness of DST models. Due to time constraints during the rebuttal period, we focused on validating the robustness of DST models in the segmentation task.
>
> We compared the performance of a dense trained fully convolutional network with a VGG16 backbone on the Pascal VOC 2012 clean dataset，then test on corrupted Pascal VOC 2012 (generated following the ImageCorruption library [1], which includes 19 types of corruption at 5 severity levels). The results are presented below. Unlike classification tasks on CIFAR or ImageNet, where the VGG16 backbone is typically pretrained on ImageNet, we first prune the backbone, then perform sparse fine-tuning on the training set of the Pascal VOC 2012 clean dataset, and finally test it on both the clean and corrupted versions of the Pascal VOC 2012 test set.
>
> The results show that DST models are **not only robust in image and video classification but also in image segmentation tasks**.
>
> Table. Clean metric (%) on Pascal VOC 2012 and  Robustness metric (%) on corrupted Pascal VOC 2012.
>
> |Metric  |  Dense training   | DST (s=0.2) | DST (s=0.1) |
> | -------- | --- | --------- | -------------- |
> | clean pixAcc ($\uparrow$)    |  85.65   |    85.63    |       **85.66**       |
> |  clean mIoU ($\uparrow$)     |  46.34   |    46.69     |       **47.10**         |
> | robustness pixAcc ($\uparrow$)    |  76.80   |    77.30     |       **77.39**        |
> | robustness mIoU ($\uparrow$) |  22.60   |    22.73     |      **23.09**      |
>
>
> [1] https://github.com/bethgelab/imagecorruptions

---

> > ### Comment · Reviewer_ncJc · 2024-11-26
> > **Thank you for the response**
> >
> > I thank the authors' response and the additional experiments on the segmentation task. It is not easy to finish those experiments in such a short period. I will increase my score.

---

> > > ### Author Response · Authors · 2024-11-26
> > >
> > > Dear Reviewer ncJc, we sincerely appreciate your constructive suggestions and thank you for your positive support!

---

### Official Review · Reviewer_EZM6 · 2024-11-03

**Soundness:** 2
**Presentation:** 3
**Contribution:** 3
**Rating:** 6
**Confidence:** 4

**Summary:**

This paper explores whether Dynamic Sparse Training (DST) results in more OOD Robust Models than Dense Training.
To this extent, the paper reports its findings using various DST methods over multiple architectures and different datasets.
The findings are consistent across experiments that DST methods lead to more OOD robust models than Dense Training.

**Strengths:**

The structure of the paper is quite different from usual submissions and is a welcome change, the style of having research questions for section headings and related experiments, findings, and discussion in those sections is interesting and keeps the reader engaged.

**Weaknesses:**

**W1- Lack of Proper Literature Review.**
The biggest weakness of this work is that it only analyzes its findings from the narrow lens of "Dynamic Sparse Training" because almost all of the questions asked in this work have been asked before from a larger context of Deep Neural Networks and Sparse Deep Neural Networks and have been successfully answered.
The related work section of this paper is very poorly written and researched.
Had significant effort been invested into finding true related works done by the community this paper would have been significantly different.
For example, in section 5.1 the paper attempts to find a correlation between inherent properties of dynamic sparse training and increased OOD robustness, however in their work [1] already showed that sparse models can be very OOD Robust, and while this proposed work on an average looks only up to 50% sparsity, [1] showed that 90% sparse models can be more robust than their dense counterparts. While [1] does not specifically look into DST, the overall observations from [1] and this work are not very dissimilar, and thus maybe there is more at play here than "inherent properties of DST", that is these observations are not unique to DST and have been explored before and shown that models can be trained to have **"reduced connectivity to specific features that are determined to be less crucial"**.

Another example is in section 5.2 the paper does an analysis using a high-pass filter and a low-pass filter. Here the conclusion in the paper is that low-frequency information is more important, high-frequency information is less important and DST methods learn some kind of inherent regularisation to focus on "more information i.e. low frequencies" and thus are more robust.
However, the claim that "low-frequency information" is "more important" information and "high-frequency information" is "less important" has not been supported by studies from this paper or from any other paper cited here.
This is essential because another exists to explain the observations that have nothing to do with "less important" and "more important" but finds its basis in signal theory and is therefore more plausible. This explanation is provided by [2] and [3] that show using only "low-frequency information" one can avoid aliasing and avoiding aliasing leads to a reduction in spectral artifacts which leads to increased OOD robustness. So, maybe DST is inherently leading to less aliasing due to the sparsity and is therefore increasing OOD Robustness.
Therefore, despite the experiments in the paper, the conclusions presented in the paper do not hold without doubt as other plausible explanations exist.

W2 - **Poorly Written.**
While the structure of the paper is quite good, the content is very poorly written and has many mistakes. Just to prove the point I will point out a few, so that similar mistakes can be avoided in the future.

1. Figure 7 subcaptions are wrong, (a) has been written as (c).
2. Math variables are not defined, example before eqn 1, it is written M $\in \mathcal{M}$, however $\mathcal{M}$ has not been defined. And in Preliminaries 3.1, $p_{tr}$ and $p_{te}$ are decomposed, but no reason has been given for it, a reason should be provided for doing an operation such as decomposition, moreover there exists $i$=1 in subscripts but $i$ is never defined.


W3- **Missing Important citations**
While I have covered some of the missing important related works in W1, it is paramount to note the missing citation for [4]. The frequency properties of common corruptions are referenced multiple times in this paper however these properties are not general knowledge but findings from [4] and they must be cited for their work.

W4 - **Outdated Data Augmentation techniques used in Appendix A.6**
MixUp is an old data augmentation technique, significantly newer data augmentation techniques exist now. Some of the most common ones while not very new are AugMix and DeepAugment [5], it might be prudent to rather (or additionally) ablate with these in in Appendix A.6.


**References**

[1] Hoffmann, Jasper, et al. "Towards improving robustness of compressed cnns." ICML Workshop on Uncertainty and Robustness in Deep Learning (UDL). Vol. 2. 2021.

[2] Grabinski, Julia, et al. "Frequencylowcut pooling-plug and play against catastrophic overfitting." European Conference on Computer Vision. Cham: Springer Nature Switzerland, 2022.

[3] Grabinski, Julia, Janis Keuper, and Margret Keuper. "Fix your downsampling ASAP! Be natively more robust via Aliasing and Spectral Artifact free Pooling." arXiv preprint arXiv:2307.09804 (2023).

[4] Yin, Dong, et al. "A Fourier perspective on model robustness in computer vision." Advances in Neural Information Processing Systems 32 (2019).

[5] Hendrycks, Dan, et al. "The many faces of robustness: A critical analysis of out-of-distribution generalization." Proceedings of the IEEE/CVF international conference on computer vision. 2021.

**Questions:**

**Q1- Section 3.1**

I fail to understand the significance of $p_{tr}(y|x)=p_{te}(y|x)$, is it supposed to mean that training and test samples are iid? If yes, then why is mentioned explicitly? What is the significance of explicitly mentioning this detail?

**Q2 - Appendix A.3.2**

To the best of my current understanding, the FLOPs calculated are mere estimations, right? These are not real gains in FLOPs that are materialized by the hardware infrastructure of today? If so, then please mention that in Appendix A.3.2, otherwise, it misleads the reader.

---

> ### Author Response · Authors · 2024-11-24
> **To Reviewer EZM6**
>
> We appreciate your positive feedback on the structure and presentation style of this paper. However, we respectfully disagree with most of the opinions expressed in your comments. Below, we address your comments one by one.
>
> > **Comment1:** W1- Lack of Proper Literature Review. The biggest weakness of this work is that it only analyzes its findings from the narrow lens of "Dynamic Sparse Training" because almost all of the questions asked in this work have been asked before from a larger context of Deep Neural Networks and Sparse Deep Neural Networks and have been successfully answered. The related work section of this paper is very poorly written and researched. Had significant effort been invested into finding true related works done by the community this paper would have been significantly different. For example, in section 5.1 the paper attempts to find a correlation between inherent properties of dynamic sparse training and increased OOD robustness, however in their work [1] already showed that sparse models can be very OOD Robust, and while this proposed work on an average looks only up to 50% sparsity, [1] showed that 90% sparse models can be more robust than their dense counterparts. While [1] does not specifically look into DST, the overall observations from [1] and this work are not very dissimilar, and thus maybe there is more at play here than "inherent properties of DST", that is these observations are not unique to DST and have been explored before and shown that models can be trained to have "reduced connectivity to specific features that are determined to be less crucial".
>
> We appreciate the reviewer's comments and would like to address each of the concerns systematically. While we value the insights, we believe there are several conceptual misunderstandings that merit clarification.
>
> **1. Motivation for Exploring Robustness in DST**
>
> The reviewer questions the value of our study of robustness in DST given prior works on pruning methods. However, the core hypothesis of our study is that DST, as a distinct paradigm from dense training, offers unique robustness benefits under corruption scenarios. While pruning research may intuitively show that reduced connectivity to specific features deemed less crucial can improve robustness, these benefits are inherently tied to dense training. In contrast, the mechanisms by which dynamic sparse training achieves robustness remain less explored, and our work aims to address this gap.
>
> Moreover, our work contributes novel insights to the community by extending the analysis beyond existing dense training paradigms. Specifically, we provide a comprehensive examination of robustness through both temporal filter-wise analysis and frequency-domain perspectives. These aspects, based on our extensive exploration of robustness against different types of corruptions, are highlighted in Sections 4 and 5 of our paper, which the reviewer seems to have overlooked.
>
> **2. Conceptual Difference Between Dynamic Sparse Training (DST) and Pruning methods**
>
> The reviewer seems to conflate dynamic sparse training (DST) with pruning methods. DST, as a recent and efficient sparse-to-sparse training paradigm, starts with a sparse neural network and simultaneously optimizes **both its weight values and topology** throughout the training process. This optimization process is the essence of dynamic sparsity, as detailed in the Introduction (line 72) and Section 3.2 of our paper. Pruning methods, on the other hand, heavily rely on a dense training either at the beginning or thoughtout training process. These two paradigms differ fundamentally in their approach and underlying principles.
>
> As a result, the findings and insights derived from pruning methods may not always apply to DST. Our work specifically focuses on analyzing robustness properties unique to DST, exploring how this sparse training paradigm differs from dense training. This focus highlights the novelty of our research and distinguishes it from existing studies on pruning methods. We hope this clarification addresses the concerns raised and underscores the importance of examining DST as a distinct training paradigm.
>
> **3. Related Work on Robustness**
>
> Starting from line 82 of the Introduction, we have included related works on traditional pruning methods and their impact on robustness. As this paper focuses on exploring dynamic sparse training (DST) as a novel paradigm for robustness against image corruption in comparison to the dense training paradigm, we believe our related work section effectively aligns with the research scope. Our aim is to expand the exploration and understanding of robustness within the context of DST, specifically in comparison to dense training, rather than reiterating the established findings from pruning methods.

---

> ### Author Response · Authors · 2024-11-24
> **To Reviewer EZM6**
>
> **4. Distinction from Work [1]**
>
> While [1] demonstrated that sparse models could achieve significant OOD robustness (even at 90% sparsity), their results relied heavily on **strong data augmentation strategies and additional JSD consistency loss, as noted in their Section 3.3 in [1]**. In contrast, our work focuses on exploring the inherent properties of DST without employing such augmentations or additional loss terms. This allows us to isolate and analyze the robustness benefits directly attributable to the dynamic sparse training paradigm, compared to dense training. Furthermore, as noted in [1], ***traditional pruned models with high compression ratios often exhibit very low OOD robustness in the absence of these augmentations (Section 3.3, [1])***. This further emphasizes that the robustness improvements observed in [1] are primarily driven by the strong data augmentation strategies and additional loss terms, rather than by sparsity itself. Therefore, we believe that comparisons under high sparsity levels with the inclusion of such augmentations or loss terms are not entirely fair.
>
> In conclusion, we respectfully disagree with the reviewer's assertion that this work lacks novelty or relevance. We hope this response clarifies our contributions and addresses the reviewer's concerns.
>
> > **Comment2:** Another example is in section 5.2 the paper does an analysis using a high-pass filter and a low-pass filter. Here the conclusion in the paper is that low-frequency information is more important, high-frequency information is less important and DST methods learn some kind of inherent regularisation to focus on "more information i.e. low frequencies" and thus are more robust. However, the claim that "low-frequency information" is "more important" information and "high-frequency information" is "less important" has not been supported by studies from this paper or from any other paper cited here. This is essential because another exists to explain the observations that have nothing to do with "less important" and "more important" but finds its basis in signal theory and is therefore more plausible. This explanation is provided by [2] and [3] that show using only "low-frequency information" one can avoid aliasing and avoiding aliasing leads to a reduction in spectral artifacts which leads to increased OOD robustness. So, maybe DST is inherently leading to less aliasing due to the sparsity and is therefore increasing OOD Robustness. Therefore, despite the experiments in the paper, the conclusions presented in the paper do not hold without doubt as other plausible explanations exist.
>
> **1.** We respectfully disagree with "the claim that 'low-frequency information' is 'more important' information and 'high-frequency information' is 'less important' has not been supported by studies from this paper or any other paper cited here."
>
> Firstly, the claim that 'low-frequency information' is 'more important' information and 'high-frequency information' is 'less important' is supported by our experiments. As discussed in observations 1 and 2 of Section 5.2, we found that attenuating low-frequency information significantly degrades classification accuracy, whereas high-frequency attenuation results in less degradation, as shown in Figure 7. This demonstrates that low-frequency information is crucial for image classification.
>
> Secondly, the claim that 'low-frequency information' is 'more important' information and 'high-frequency information' is 'less important' is supported by the paper we cited. In observation 1 and Section 5.2, we mentioned "the observations of the crucial role of low-frequency information in image recognition tasks, consistent with findings in figure 1 of [4]."
>
>
> **2.** "This explanation is provided by [2] and [3] that show using only "low-frequency information" one can avoid aliasing and avoiding aliasing leads to a reduction in spectral artifacts which leads to increased OOD robustness. So, maybe DST is inherently leading to less aliasing due to the sparsity and is therefore increasing OOD Robustness."
>
> DST relies more on 'low-frequency information,' which inherently leads to less aliasing and presents another explanation worth further exploration. Our explanation is straightforward and intuitive: if the DST model relies less on 'high-frequency information,' it will be less affected by corruptions containing more 'high-frequency information' compared to dense training, which utilizes more 'high-frequency information' for classification. Furthermore, the experimental results in Figures 3 and 4 validate that DST models are more robust to corruptions containing high-frequency information.

---

> ### Author Response · Authors · 2024-11-24
> **To Reviewer EZM6**
>
> **3.** We respectfully disagree with the assertion that the existence of other plausible explanations precludes the possibility of additional valid explanations. Scientific inquiry and analysis often allow for multiple complementary or alternative perspectives to coexist, each contributing to a deeper understanding of the phenomenon. Therefore, while other plausible explanations may exist, they do not invalidate or diminish the significance of the explanation we have presented in our work.
>
> > **Comment3:** Figure 7 subcaptions are wrong, (a) has been written as ( c ).
>
> Thanks for pointing out, we have revised in the latest version.
>
> > **Comment4:** Math variables are not defined, example before eqn 1, it is written M, however has not been defined. And in Preliminaries 3.1, and are decomposed, but no reason has been given for it, a reason should be provided for doing an operation such as decomposition, moreover there exists i=1 in subscripts but is never defined.
>
> **1.** "before eqn 1, it is written M, however has not been defined" and "there exists i=1 in subscripts but is never defined", thanks for pointing out, we have revised in the latest version.
>
> **2.** The joint probability distribution of two random variables x, y, denoted P(x,y), can be decomposed into a marginal distribution and a conditional distribution. This decomposition is a fundamental concept in probability and statistics.
>
> > **Comment5:** Missing Important citations While I have covered some of the missing important related works in W1, it is paramount to note the missing citation for [4]. The frequency properties of common corruptions are referenced multiple times in this paper however these properties are not general knowledge but findings from [4] and they must be cited for their work.
>
> Actually, We would like to clarify that this paper is already cited in our submission, **specifically in Section 5.2 twice**. We will review the all details to ensure clarity and accuracy and address any issues in the final version.
>
>
> > **Comment6:** Outdated Data Augmentation techniques used in Appendix A.6 MixUp is an old data augmentation technique, significantly newer data augmentation techniques exist now. Some of the most common ones while not very new are AugMix and DeepAugment [5], it might be prudent to rather (or additionally) ablate with these in in Appendix A.6.
>
> Thank you for your comment. While MixUp is indeed a traditional data augmentation technique, it remains widely used and effective in image tasks. The primary goal of this paper is not to propose a new state-of-the-art method but to critically doubt the robustness of dense training compared to the training paradigm of dynamic sparse training (DST) in the context of image corruptions. This provides a novel perspective on robustness analysis through DST rather than traditional dense training.
>
> In Appendix A.6, our analysis focused on exploring how data augmentation interacts with DST compared to dense training. As stated at the beginning of A.6, our findings demonstrate that combining data augmentation with DST can be highly effective in improving robustness against image corruptions, outperforming dense training. While additional ablations with newer techniques such as AugMix or DeepAugment could provide more insights, we believe these would have limited impact on the central research questions and claims of our work. We appreciate the suggestion and may consider such ablations in future work to further extend the scope of analysis.
>
>
> > **Comment7:** I fail to understand the significance of $p_{tr}(y \mid x)=p_{te}(y \mid x)$, is it supposed to mean that training and test samples are iid? If yes, then why is mentioned explicitly? What is the significance of explicitly mentioning this detail?
>
> $p_{tr}(y \mid \boldsymbol{x})=p_{te}(y \mid \boldsymbol{x})$ means that feature-label relationship consistent for training and test samples, which does not imply training and test samples are i.i.d., because $p_{tr}(\boldsymbol{x}) \neq  p_{te}(\boldsymbol{x})$, as defind in section 3.1.
>
> The difference $p_{tr}(\boldsymbol{x}) \neq p_{te}(\boldsymbol{x})$ reflects a real-world scenario where models trained on clean data encounter unseen corruptions or domain shifts in deployment. The condition $p_{tr}(y \mid \boldsymbol{x})=p_{\mathrm{te}}(y \mid \boldsymbol{x})$ ensures that test performance depends only on the model's ability to handle distribution shifts in the input $\boldsymbol{x}$ and not on mismatched label distributions.

---

> ### Author Response · Authors · 2024-11-24
> **To Reviewer EZM6**
>
> > **Comment8:** To the best of my current understanding, the FLOPs calculated are mere estimations, right? These are not real gains in FLOPs that are materialized by the hardware infrastructure of today? If so, then please mention that in Appendix A.3.2, otherwise, it misleads the reader.
>
> Thank you for your comment and pointing out your questions. Indeed, the computational costs in this work are theoretical and not fully realized gains on current hardware infrastructures. This is a common limitation in the literature on sparse training methods, as most current hardware and software systems are optimized for dense matrix operations rather than sparse ones.
>
> However, recent advancements in model sparsity are increasingly aligning with hardware and software developments to fully leverage the benefits of sparsity. For example, NVIDIA’s A100 GPU supports 2:4 sparsity as described in [1], and other innovations are making strides toward efficient sparse implementations [2,3]. Simultaneously, software libraries such as [4] are emerging to enable truly sparse network implementations.
>
> These advancements are paving the way for future deep neural networks to achieve greater efficiency in terms of computation, memory, and energy use. We have clarified the limitations of current hardware support and emphasize the estimated nature of the FLOPs in **Appendix A.3.2** in our latest version to avoid any potential misunderstanding.
>
> [1] Zhou, Aojun, et al. "Learning N: M Fine-grained Structured Sparse Neural Networks From Scratch." ICLR 2021.
>
> [2] Chen, Yu-Hsin, et al. "Eyeriss v2: A flexible accelerator for emerging deep neural networks on mobile devices." IEEE Journal on Emerging and Selected Topics in Circuits and Systems (2019).
>
> [3] Ashby, Mike, et al. "Exploiting unstructured sparsity on next-generation datacenter hardware." (2019).
>
> [4] Liu, Shiwei, et al. "Sparse evolutionary deep learning with over one million artificial neurons on commodity hardware." Neural Computing and Applications 33 (2021).
>
> Thank you for your comments. If you feel that we have addressed your concerns, we would be grateful if you could reconsider your score. Thanks again for your time and effort.

---

> ### Comment · Reviewer_EZM6 · 2024-11-26
> **Clearing misunderstandings by the review**
>
> Dear Authors,
>
> I would like to start by acknowledging that I should have done a better job highlighting the paper's strengths. My focus while writing the review was on highlighting key points, that in my opinion, would significantly improve the quality of the paper, and the experience of the reader in terms of knowledge gained. However, I now realize that highlighting the paper's strengths helps set a better tone for the review itself, something for me to improve upon for the next reviews. I assumed, given the time constraints, that focusing on the possible aspects of improving the paper might be better appreciated than highlighting the obvious facts. In hindsight, doing so has made it seem that I did not understand DST in general and it's importance. I would like to reassure the authors that this is not the case.
> I would like to state unequivocally that I acknowledge "This optimization process is the essence of dynamic sparsity, ...... Pruning methods, .....These two paradigms differ fundamentally in their approach and underlying principles. As a result, the findings and insights derived from pruning methods **may not always apply to DST.** Our work specifically focuses on analyzing robustness properties unique to DST, **exploring how this sparse training paradigm differs from dense training.** This focus highlights the novelty of our research and distinguishes it from existing studies on pruning methods. ... importance of examining DST as a distinct training paradigm."
> This fact is not amiss to me, and again, something I should have highlighted in the paper's strengths.
>
>
> Next, before I begin replying to the current answers, I would like to highlight that in my review, I took time and effort to even read through the appendix and suggest improvements to specific sections in the appendix, yes, I might have missed one citation, but if I took the time to read even the appendix in detail, I am certain that I did not miss out on reading two entire sections of the paper, and thus suggesting that "are highlighted in Sections 4 and 5 of our paper, **which the reviewer seems to have overlooked.**" is disappointing to read at the least, if not offensive. However, as mentioned in the para above, I might not have set the right tone by "focusing on aspects of improvement" and not investing effort in highlighting the strengths, thus, for now, I choose to overlook this.
>
> Firstly, "The joint probability distribution of two random variables x, y, denoted P(x,y), can be decomposed into a marginal distribution and a conditional distribution. This decomposition is a fundamental concept in probability and statistics."
>
> **Answer**: I believe my question has been misunderstood. I am well aware of fundamental concepts in probability. My question is not about the concept itself but why it is explicitly been stated when it is not really used ever again. So, to the best of my understanding, the entire paper has used simple language to explain theoretical concepts of robustness. However, in the preliminaries, there are some math foundations laid out, just to say a very simple thing, "feature-label relationship is consistent for training and test samples". This statement can be easily made with a few words of text, and thus using 6 lines of equations + text is inefficient. While I appreciate mathematical concepts being used in a paper, I personally appreciate consistency more. The paper currently explicitly introduces the distributions $p_{tr}$, and $p_{te}$ in the preliminaries of the paper, and then in the very next sentence switches back to text. If variables $p_{tr}$ and $p_{te}$ are being introduced, then they should also be used, by explicitly mentioning that here during training, the training datasets of CIFAR-10, CIFAR-100, Tiny Imagenet, and ImageNet is $p_{tr}$ when CIFAR10-C, CIFAR100-C, TinyImageNet-C and ImageNet-C/$\bar{\text{C}}$/3DCC is $p_{te}$, respectively. If these variables are not used, then they have been introduced for no reason. And, thus maybe a better approach would be to not use them at all and simply write in text that  "feature-label relationship consistent for training and test samples".

---

> ### Comment · Reviewer_EZM6 · 2024-11-26
> **[contd.] Clearing misunderstandings by the review**
>
> Secondly, I believe a lot of my points of improvement have been taken out of context. I'm not denying the existence of results in the paper, and neither am I stating that two possible explanations cannot exist for that same behavior. As mentioned at the start of my review, "The biggest weakness of this work is that it only analyzes its findings from the narrow lens of "Dynamic Sparse Training" because almost all of the questions asked in this work have been asked before from a larger context of Deep Neural Networks and Sparse Deep Neural Networks and have been successfully answered.". This does not mean that the findings for DST are not novel, but it rather means that at the end of the day, DST and also a Deep Learning Approach, and if the possibility exists to broaden one's scope and look at findings with a broader view then missing this opportunity would be a great loss. For example, in my statement, "paper does an analysis using a high-pass filter and a low-pass filter. Here the conclusion in the paper is that low-frequency information is more important, high-frequency information is less important and DST methods learn some kind of inherent regularisation to focus on "more information i.e. low frequencies" and thus are more robust. However, the claim that "low-frequency information" is "more important" information and "high-frequency information" is "less important" has not been supported by studies from this paper or from any other paper cited here", this is indeed true! The experiments currently only show the correlations between attenuating either low or high-frequency data in the input and respective classification accuracies. I sincerely urge reading the papers I cited, as these works present a more plausible cause for this behavior than "DST methods learn some kind of inherent regularisation to focus on "more information i.e. low frequencies" and thus are more robust." Currently there indeed exist no experiments in this paper to truly show that low frequencies mean more information which leads to robustness. DST could simply be learning regularization to focus on low-frequency information, leading to reduced aliasing, leading to learning better feature representations, leading to "improved robustness".
>
> Similarly, for the other points raised in my review, yes, DSTs are unique, but they are also a Deep Learning Method. And, yes the unique behavior of DST methods is very interesting, but what highlights these unique behaviors when compared to other DL approaches is the fact that there are a lot of similarities as well.
> That is, in light of the similarities, there exist few uniquenesses, and thus these unique behaviors are interesting and worth looking at. This is what I tried to put forward with my review: "Yes, there are unique behaviors of DST worth looking at, and why these behaviors are different and interesting, is because for behavior A we find a parallel in other DL approaches [citations for this], and for behavior B, we also find plausible explanations from other DL approaches [citations for these], however, for behavior C we find no parallels in DL approaches, in fact these behaviors are contrary to known behaviors observed in other DL approaches [citations for those], and so on."
>
> In my opinion, presenting the findings of the proposed method in this way would provide the readers with a more wholesome view of the work. And, I suggest this since the title is "Dynamic Sparse Training versus Dense Training", I believe is it important to point out possible similarities while pointing out differences.
>
> My initial rating of "strong reject" existed due to inaccuracies in the math (undefined variable), and I could not give a paper with undefined variables any other recommendation. Since that is now fixed, I have revised my evaluation to better reflect my position.
> In light of this "clearing of misunderstandings" I have tried to provide, I urge the authors to revisit my original review and attempt to revise the answers and approach to the review.
>
> Best Regards
>
> Reviewer EZM6

---

> ### Author Response · Authors · 2024-11-28
>
> Dear Reviewer EZM6, we sincerely thank you for re-evaluating our paper and for your time and effort in reviewing and actively engaging with our responses. We greatly value the opportunity for further discussion. However, we respectfully hold some differing opinions regarding certain points raised in your comments, and we address these concerns below, point by point.
>
> > "I would like to start by acknowledging that I should have done a better job highlighting the paper's strengths. My focus while writing the review was on highlighting key points, that in my opinion, would significantly improve the quality of the paper, and the experience of the reader in terms of knowledge gained. However, I now realize that highlighting the paper's strengths helps set a better tone for the review itself, something for me to improve upon for the next reviews. I assumed, given the time constraints, that focusing on the possible aspects of improving the paper might be better appreciated than highlighting the obvious facts."
>
> We sincerely thank you for the effort you have invested in improving the quality of our paper. We agree that overlooking the strengths and contributions of a paper can unintentionally undermine the authors' efforts and the value their work brings to the community. For better evaluation, highlighting not only the weaknesses but also the strengths of a paper helps us identify areas for improvement and provides a fair and respectful evaluation of the work. We genuinely appreciate your thoughtful reflection on ways to further enhance the reviewing process.
>
>
>
> > "I would like to state unequivocally that I acknowledge "This optimization process is the essence of dynamic sparsity, ...... Pruning methods, .....These two paradigms differ fundamentally in their approach and underlying principles. As a result, the findings and insights derived from pruning methods may not always apply to DST. Our work specifically focuses on analyzing robustness properties unique to DST, exploring how this sparse training paradigm differs from dense training. This focus highlights the novelty of our research and distinguishes it from existing studies on pruning methods. ... importance of examining DST as a distinct training paradigm." This fact is not amiss to me, and again, something I should have highlighted in the paper's strengths."
>
> We appreciate your acknowledgment of the importance of dynamic sparsity (DST) and its distinction from pruning methods. We are also grateful for your recognition of strengths of our paper.
>
> > ".....I took time and effort to even read through the appendix and suggest improvements to specific sections in the appendix, yes, I might have missed one citation, but if I took the time to read even the appendix in detail, I am certain that I did not miss out on reading two entire sections of the paper,and thus suggesting that "are highlighted in Sections 4 and 5 of our paper, which the reviewer seems to have overlooked." is disappointing to read at the least, if not offensive."
>
> We sincerely appreciate the time and effort you dedicated to thoroughly reviewing our paper, including the appendix, and for providing detailed and thoughtful feedback. Regarding the missed citation mentioned in your comment, we want to clarify that our intention was never to undermine your effort but to ensure that nothing critical was overlooked. If our wording, particularly the phrase "are highlighted in Sections 4 and 5 of our paper, which the reviewer seems to have overlooked," caused any misunderstanding or led to disappointment or offense, we sincerely apologize. Please rest assured that we deeply respect and value the time and effort you have invested in reviewing our work.

---

> ### Author Response · Authors · 2024-11-28
>
> > My question is not about the concept itself but why it is explicitly been stated when it is not really used ever again. .... thus maybe a better approach would be to not use them at all and simply write in text that "feature-label relationship consistent for training and test samples
>
> Thank you for your suggestion regarding the use of mathematical notation in the preliminaries section.
>
> Our original intent in introducing the problem definition mathematically was to provide a precise and unambiguous explanation of the relationship between the training and test datasets, emphasizing that the feature-label relationship $p(y \mid \mathbf{x})$ remains consistent while the marginal feature distributions $p(\mathbf{x})$ differ. However, we recognize that this could have been communicated more efficiently and aligned more closely with the overall narrative style of the paper.
>
> To address your feedback and improve the paper, we have made the following revisions for section 3.1 PROBLEM DEFINITION:
>
> *Our goal is to evaluate the model's robustness in scenarios where the test inputs are corrupted relative to the benchmarked training datasets, (i.e., $p_{\text {tr }}(\boldsymbol{x}) \neq p_{\text {te}}(\boldsymbol{x})$ while the conditional input-label relationship remains consistent (i.e., $\left.p_{\operatorname{tr}}(y \mid \boldsymbol{x})=p_{\mathrm{te}}(y \mid \boldsymbol{x}\right)$). To achieve this, we utilize standard image benchmark datasets such as CIFAR10, CIFAR100,  TinyImageNet and ImageNet for training. Then, we evaluate the performance (robustness accuracy) of the models on the corresponding corruption benchmark datasets, including CIFAR10-C, CIFAR100-C, TinyImageNet-C and ImageNet-C/$\bar{{C}}$/3DCC.*
>
>
> > I'm not denying the existence of results in the paper, and neither am I stating that two possible explanations cannot exist for that same behavior.
>
> Thank you for acknowledging the validity of the results presented in our paper. We appreciate your recognition and your openness to the possibility of multiple explanations for the observed behavior.
>
>
> > "low-frequency information" is "more important" information and "high-frequency information" is "less important" has not been supported by studies from this paper or from any other paper cited here", this is indeed true! The experiments currently only show the correlations between attenuating either low or high-frequency data in the input and respective classification accuracies.
>
> It is worth clarifying that our experiments demonstrate clear correlations between attenuating specific frequency and classification accuracy. This analysis is motivated by the empirical results in Section 4.2, where we observed that DST models are surprisingly more effective at countering corruptions rich in high-frequency information compared to their dense training counterparts.
>
> We hypothesize that this advantage arises because DST models inherently de-emphasize high-frequency components, as discussed at the end of Section 5.1 (Lines 412–414). This behavior significantly enhances their robustness, **particularly against corruptions dominated by high-frequency noise**, thereby leading to improved overall robustness compared to dense training. These findings not only highlight the correlations between frequency attenuation and classification accuracy but also provide substantial evidence supporting the relationship between the importance of low/high-frequency information and robustness in DST models.

---

> ### Author Response · Authors · 2024-11-28
>
> > DST methods learn some kind of inherent regularisation to focus on "more information i.e. low frequencies" and thus are more robust. .... there indeed exist no experiments in this paper to truly show that low frequencies mean more information which leads to robustness. DST could simply be learning regularization to focus on low-frequency information, leading to reduced aliasing, leading to learning better feature representations, leading to "improved robustness.
>
> We believe there is a logical misunderstanding in equating "low frequency is more important" with "then more robust." In our work, we find that DST models inherently de-emphasize high-frequency components. This behavior enhances their robustness, particularly against corruptions dominated by high-frequency noise, thereby leading to improved overall robustness compared to dense training.
>
> From your recommended papers [2,3], high-frequency components are often associated with aliasing, which can reduce overall robustness. In contrast, our work shows that DST models inherently de-emphasize high-frequency components, leading to improved robustness against high-frequency corruptions. In this sense, the findings are aligned. **However, the key difference lies in our analysis, which is specifically driven by results from different corruption types, offering a distinct perspective.**
>
> Regarding your observation that "DST could simply be learning regularization to focus on low-frequency information, leading to reduced aliasing, better feature representations, and improved robustness," we sincerely appreciate this insightful suggestion. We agree that this is a promising avenue for further investigation, and we have added a discussion on this topic at the end of Section 5.3 to inspire future studies in this direction.
>
>
> > This does not mean that the findings for DST are not novel, but it rather means that at the end of the day, DST and also a Deep Learning Approach, and if the possibility exists to broaden one's scope and look at findings with a broader view then missing this opportunity would be a great loss. I sincerely urge reading the papers I cited, as these works present a more plausible cause for this behavior than "DST methods learn some kind of inherent regularisation to focus on "more information i.e. low frequencies" and thus are more robust."
>
> Thank you for acknowledging the novelty of our findings and for suggesting a broader perspective and related works. As discuss before, high-frequency components often lead to aliasing, reducing robustness [2,3]. In contrast, our work demonstrates that DST models inherently de-emphasize high-frequency components, enhancing robustness especially against high-frequency corruptions. Our analysis is uniquely driven by results across different corruption types, offering a distinct perspective. We agree that papers [2,3] present promising avenues for further exploration, and **we have added a discussion on this topic at the end of Section 5.3 to inspire future studies.**
>
> > Similarly, for the other points raised in my review, yes, DSTs are unique, but they are also a Deep Learning Method. And, yes the unique behavior of DST methods is very interesting, but what highlights these unique behaviors when compared to other DL approaches is the fact that there are a lot of similarities as well. That is, in light of the similarities, there exist few uniquenesses, and thus these unique behaviors are interesting and worth looking at. This is what I tried to put forward with my review: "Yes, there are unique behaviors of DST worth looking at, and why these behaviors are different and interesting, ......., I believe is it important to point out possible similarities while pointing out differences.
>
> Thank you for your suggestion. While the title "Dynamic Sparse Training versus Dense Training" emphasizes a comparison, our primary aim in this work is to focus on the differences between these two paradigms, specifically in the context of corruption robustness. The motivation stems from our findings that DST and dense training exhibit distinct behaviors when exposed to different types of corruptions. This motivated our deeper spatial and spectral analyses, which highlight these differences.
>
> We agree that exploring potential similarities could add value; however, we argue that such an analysis, while complementary, would detract from the primary focus of our study. A detailed investigation into the similarities between DST and dense training would be better suited for a separate and dedicated study. By centering on the distinctions, our work provides clear and actionable insights into why DST emerges as the unexpected winner in image corruption robustness, as highlighted in the title, thereby aligning with the central aim of this paper.
>
> Best regards,
>
> Authors

---

> ### Comment · Reviewer_EZM6 · 2024-11-28
>
> Dear Authors,
>
> Thank you for addressing some of my concerns.
> However, three concerns remain, firstly now in section 3.1 the variables $n_{tr}$ and $n_{te}$ are not defined, it should be mentioned that these are all training samples and test samples respectively.
>
> Secondly, Table 3 does not look good, it is not consistent with the other tables in the paper.
>
> Lastly, I respectfully disagree with "however, we argue that such an analysis, while complementary, would detract from the primary focus of our study. A detailed investigation into the similarities between DST and dense training would be better suited for a separate and dedicated study." I hold the opinion that when a comparative analysis is made, one that is advertised in the title itself, a study on both similarities and differences is paramount to provide a better understanding. This work primarily focuses on the latter.
>
> Nonetheless, since some of my concerns have been addressed, and since now the paper provides a slightly more holistic view of DST and Deep Learning by improving on the statements and arguments, I will raise my score to borderline accept, hoping that my first two concerns would be addressed in the camera-ready version.
>
> In light my these remaining concerns, I do not hold the opinion that this work is currently ready to be completely endorsed by me. I would rely on the authors' answers to these concerns and the views of my fellow reviewers to make a final recommendation.
>
> Best Regards
>
> Reviewer EZM6

---

> ### Author Response · Authors · 2024-11-30
>
> Dear Reviewer EZM6,
>
> Thank you once again for your valuable comments and positive support for our work.
>
> > **Comment1:** firstly now in section 3.1 the variables $n_{tr}$ and $n_{te}$ are not defined, it should be mentioned that these are all training samples and test samples respectively. Secondly, Table 3 does not look good, it is not consistent with the other tables in the paper.
>
> Thank you very much for pointing out these detailed issues. We sincerely appreciate your careful and thorough review. Specifically, we have clarified that "the variables $n_{tr}$ and $n_{te}$ indicate the total number of training and test samples, respectively." Regarding Table 3, we have ensured its formatting is consistent with the other tables in the paper. While we are currently unable to update the submission version, we will carefully address and revise these points in the camera-ready version of the paper.
>
> > **Comment2:** Lastly, I respectfully disagree with "however, we argue that such an analysis, while complementary, would detract from the primary focus of our study. A detailed investigation into the similarities between DST and dense training would be better suited for a separate and dedicated study." I hold the opinion that when a comparative analysis is made, one that is advertised in the title itself, a study on both similarities and differences is paramount to provide a better understanding. This work primarily focuses on the latter.
>
> Thank you for your valuable suggestions and advice. While the breadth of compressive similarity study extends far beyond the scope of this paper—which has a focused objective—and would require a series of papers and a broader community effort, we agree that a brief discussion of both the similarities and differences between DST and dense training would provide a more holistic understanding and better motivate our topic. Therefore, we propose revising one of the paragraphs in our submission (line 80~81) as follows:
>
> *However, DST shares many commonalities with dense training in its versatility and foundational strategies. Like dense training, DST can be seamlessly applied to various learning paradigms, including continual learning [1], reinforcement learning [2][3][4], and diverse tasks such as feature selection [5][6], time series classification [7], medical image analysis [8], and network architecture design [9]. Furthermore, both approaches are compatible with a wide range of architectures, including convolutional and transformer-based models [10]. DST also incorporates key foundational training strategies intrinsic to dense training, such as dropout, weight decay, and optimizers, although identifying the optimal hyperparameters could unlock DST's full potential [11]. Despite these similarities, evaluations in the literature have primarily been limited to clean benchmarks, where the training and test sets share identical distributions. What sets DST apart, however, is its use of hard sparsity, in addition to the soft sparsity promoted by weight decay or other regularization techniques, inducing additional regularization. This raises an intuitive research question: Is somehow DST implicitly improving model robustness, and no one knew about this yet?*
>
> We hope to have addressed all your concerns thoroughly. If you have any further suggestions or questions, please feel free to share them with us. We are always open to constructive discussions.
>
> [1] Ghada Sokar, et al. Avoiding forgetting and allowing forward transfer in continual learning via sparse networks. ECML PKDD 2022.
>
> [2] Laura Graesser, et al. The state of sparse training in deep reinforcement learning. ICML, 2022.
>
> [3] Yiqin Tan, et al. RLx2: Training a sparse deep reinforcement learning model from scratch. ICLR 2023.
>
> [4] G Sokar, et al. Dynamic Sparse Training for Deep Reinforcement Learning. IJCAI-ECAI 2022.
>
> [5] Ghada Sokar, et al. Where to pay attention in sparse training for feature selection? NeurIPS 2022.
>
> [6] Zahra Atashgahi, et al. Quick and robust feature selection: the strength of energy-efficient sparse training for autoencoders. Machine Learning.
>
> [7] Qiao Xiao, et al. Dynamic sparse network for time series classification: Learning what to “see”. NeurIPS 2022.
>
> [8] Boqian Wu, et al. E2ENet: Dynamic Sparse Feature Fusion for Accurate and Efficient 3D Medical Image Segmentation. NeurIPS 2024.
>
> [9] Shiwei Liu, et al. More convnets in the 2020s: Scaling up kernels beyond 51x51 using sparsity. ICLR 2023.
>
> [10] Tianlong Chen, et al. Chasing sparsity in vision transformers: An end-to-end exploration. NeurIPS 2021.
>
> [11] Nolan Simran Dey, et al. Sparse maximal update parameterization: A holistic approach to sparse training dynamics. NeurIPS 2024.
>
> Best regards,
>
> Authors

---

### Official Review · Reviewer_MR2K · 2024-11-04

**Soundness:** 2
**Presentation:** 3
**Contribution:** 2
**Rating:** 6
**Confidence:** 4

**Summary:**

The following paper explores the effect of conducting Dynamic Sparse Training (DST) within a neural network towards its robustness to adversarial samples. First, the paper introduces the Dynamic Sparsity Corruption Hypothesis, which postulates that performing DST on artificial neural networks at a low sparsity regime improves its robustness against image and video corruption tasks compared to dense training. Next, The paper proceeds to validate this hypothesis by conducting experiments on various DST algorithms under different deep learning architectures, data modalities, and data corruption scenarios and their severity and comparing their robustness and accuracy with dense training. Then, the authors analyze spectral and spatial domains to show that DST works as an implicit regularizer for neural networks by reducing its attention to high-frequency components or less meaningful features.

**Strengths:**

1. Writing is easy to follow, and the content organization is clear, making the paper easy to read and comprehend.
2. A comprehensive set of evaluations is conducted to show Dynamic Sparse Training (DST) achieves better performance in terms of robustness accuracy than its dense counterparts.
3. While this paper does not propose a new method that improves state-of-the-art performance at a specific benchmark, it does provide new insights into the unexpected benefit of performing DST on image and video corruption tasks in favor of dense training on varying neural network architectures.

**Weaknesses:**

I hope the authors can provide revision for the paper based on the following points:

1. There is no detailed explanation for several experiments conducted in the main paper. Such details are as follows:
- The experiment was conducted on the ImageNet-3DCC dataset in Section 4.3. One needs to find Table 3 at the end of page 10 to see the model's sparsity ratio employed for the experiment. Even then, there is no detailed explanation of how to train the model for experiments at both the ImageNet-3DCC dataset and the UCF101 dataset for video representation.
- There are lack of information regarding the details of the experiment conducted in Section 6.1. It only briefly mentions that it uses an MLP model without explicitly mentioning its detailed architecture. In addition, I fail to find details/recipes on how you train the model to achieve results in Table 2.

2. I also don't get why the sparsity ratio differs for experiments in different Sections (namely Section 4.1 and the rest of the subsections in Section 4). Namely, in section 4.1, the authors are trying to evaluate the robustness accuracy in various sparsity ratios ranging from 0.3 to 0.7, whereas the rest of the experiments are conducted at a low sparsity regime of 0.1. I also want to ask the same question regarding the motivation in setting the initial density of GraNet to be 0.8 and the motivation for setting the soft memory bound for MEST to 10% of the target density. I hope the author can address the reasoning for doing so in their rebuttal.

3. The main claims that this paper is trying to address are a bit problematic for me. For example, the DSCR hypothesis mentions a "low sparsity regime", but the authors fail to quantify which range of values would be suitable to be defined as a "low" sparsity regime and one that is not in principle. This issue, I think, connects with issue 2 pointed out above, which makes the sparsity constraint $S$ for different experiments about different datasets different.

4. (Minor) These are things that might need to be revised in the paper and the appendix:
- "Y-axis in Figure 3" should be "x-axis in Figure 3". [Section 4.2]
- **ImageNet-C** -> "generated from Tiny-ImageNet" should be "generated from ImageNet" [Appendix A.2]
- **CIFAR-100** -> ReNet-34 should be ResNet-34 [Appendix A.3.2]
- "(c)" should be "(a)" [Figure 7 Right-Top] (I think it's better to put (a), (b), and (c) below the bottom figure and then explain the right figure without mentioning the top and bottom part so that (d), (e), and (f) don't get wasted there).

**Questions:**

1. In Figure 3 within the main paper, what do these five bars represent in each corruption type? Please indicate this in a legend within one of the graphs for clarity.
2. Regarding analyses conducted in Section 5.1 and Section 5.2, what are the motivations for using the random regrowth procedure for both MEST and GraNet instead of their gradient-based regrowth variants for comparison in Figure 5 and Figure 7? I am asking because the results displayed on the paper (Figure 3, Figure 4, Table 1, and Table 3) use the latter instead of the former.
3. What causes each DST method to achieve worse performance in a particular type of corruption (i.e., single_freq (single frequency greyscale) in ImageNet-$\bar{\mathbf{C}}$)?

---

> ### Author Response · Authors · 2024-11-24
> **To Reviewer MR2K**
>
> We sincerely thank you for your time and effort in reviewing our paper. We are pleased to hear that you find our work thorough, well-structured, and insightful. We address your comments below.
>
> > **Comment1:** The experiment was conducted on the ImageNet-3DCC dataset in Section 4.3. One needs to find Table 3 at the end of page 10 to see the model's sparsity ratio employed for the experiment. Even then, there is no detailed explanation of how to train the model for experiments at both the ImageNet-3DCC dataset and the UCF101 dataset for video representation.
> There are lack of information regarding the details of the experiment conducted in Section 6.1. It only briefly mentions that it uses an MLP model without explicitly mentioning its detailed architecture. In addition, I fail to find details/recipes on how you train the model to achieve results in Table 2.
>
>
> **1. ImageNet-3DCC Dataset (Section 4.3):**
> The model evaluated on ImageNet-3DCC dataset is identical to the ResNet50 training on ImageNet dataset. Detailed implementation details can be found in Appendix A.3.1.
>
> **2. UCF101 Dataset:**
> For training on the UCF101 dataset, as mentioned on line 330, we followed the settings described in [1]. The training setup is based on the code available at https://github.com/Maddy12/ActionRecognitionRobustnessEval. Specifically, the model was trained for 80 epochs with an initial learning rate of 0.1 using the SGD optimizer, with a weight decay of 1e-3.
>
> **3. Section 6.1 Experimental Setup:**
> The experimental setup for Section 6.1 follows the configuration described in [2]. The implementation is based on the code available at https://github.com/biomedical-cybernetics/Cannistraci-Hebb-training. The MLP model used has hidden dimensions of [728, 728], with a ReLU activation applied after each layer. The model was trained for 100 epochs with an initial learning rate of 0.025 using the SGD optimizer and a weight decay of 5e-4.
>
> We appreciate you pointing this out and will include these details in **Appendix A.3.1 and Table 4** for better clarity. Thank you for your constructive feedback.
>
> > **Comment2:** I also don't get why the sparsity ratio differs for experiments in different Sections (namely Section 4.1 and the rest of the subsections in Section 4). Namely, in section 4.1, the authors are trying to evaluate the robustness accuracy in various sparsity ratios ranging from 0.3 to 0.7, whereas the rest of the experiments are conducted at a low sparsity regime of 0.1. I also want to ask the same question regarding the motivation in setting the initial density of GraNet to be 0.8 and the motivation for setting the soft memory bound for MEST to 10% of the target density. I hope the author can address the reasoning for doing so in their rebuttal.
>
> Thank you for sharing these concerns. We address your points below:
>
> **In section 4.1, the authors are trying to evaluate the robustness accuracy in various sparsity ratios ranging from 0.3 to 0.7, whereas the rest of the experiments are conducted at a low sparsity regime of 0.1.**
>
> For small-scale datasets such as CIFAR and TinyImageNet, empirical evidence suggests that the model generally achieves decent performance on clean datasets even at high sparsity levels [3]. This is because the model's capacity is sufficient to learn well under these conditions. Motivated by this, we extended our exploration to higher sparsity levels, ranging from 0.3 to 0.7, in Section 4.1, aiming to comprehensively evaluate robustness performance across a broader spectrum of sparsity ratios.
>
> In contrast, for large-scale datasets like ImageNet1K, models often exhibit a significant decline in capacity at higher sparsity levels, leading to notable performance degradation even on clean datasets (as demonstrated in [4]). To address this and ensure an effective evaluation of robustness, we aimed to minimize the impact on the model's capacity. Consequently, we adopted a lower sparsity level (e.g., 0.1) in these experiments, enabling a fairer and more reliable comparison of robustness between dynamic sparse training and dense training.
>
> **Why initial density of GraNet to be 0.8 and the motivation for setting the soft memory bound for MEST to 10% of the target density.**
>
> The goal of GraNet is to gradually prune the model during dynamic sparse training, starting with a higher initial density to retain sufficient capacity for learning in the early stages. Therefore, the initial sparsity is set much lower than the final sparsity. Specifically, we chose a consistent initial density of 0.8 (sparsity 0.2) for final sparsity ratios ranging from 0.3 to 0.7, allowing for fair comparisons between different target sparsity levels.
>
> MEST assumes a soft memory bound (approximately 10% more density, which is the default setting in the original MEST) during sparse training, meaning the sparsity during training is slightly higher than the target sparsity. This design is motivated by practical considerations.

---

> ### Author Response · Authors · 2024-11-24
> **To Reviewer MR2K**
>
> [1] Schiappa, Madeline Chantry, et al. "A large-scale robustness analysis of video action recognition models." CVPR 2023.
>
> [2] Zhang, Yingtao, et al. "Epitopological learning and Cannistraci-Hebb network shape intelligence brain-inspired theory for ultra-sparse advantage in deep learning." ICLR 2024.
>
> [3] Nowak, Aleksandra, et al. "Fantastic weights and how to find them: Where to prune in dynamic sparse training." NeurIPS 2024.
>
> [4] Evci, Utku, et al. "Rigging the lottery: Making all tickets winners." ICML 2020.
>
> > **Comment3:** The main claims that this paper is trying to address are a bit problematic for me. For example, the DSCR hypothesis mentions a "low sparsity regime", but the authors fail to quantify which range of values would be suitable to be defined as a "low" sparsity regime and one that is not in principle. This issue, I think, connects with issue 2 pointed out above, which makes the sparsity constraint for different experiments about different datasets different.
>
> Thank you for your insightful comment. Indeed, the primary focus of our paper is on the "low sparsity regime." This choice stems from our motivation to study robustness in sparse training while maintaining model capacity. Such a focus ensures **meaningful and fair evaluations** of robustness under sparsity constraints.
>
> However, for small-scale datasets such as CIFAR and Fashion-MNIST, we have also conducted exploratory studies across various sparsity levels, as shown in Table 3 and Figure 2. The rationale is that for these datasets, even at higher sparsity levels, the model often retains sufficient capacity to learn effectively, resulting in decent clean performance, as demonstrated in [1]. This exploration aligns with our broader objective of understanding robustness under different sparsity levels while ensuring that the model's capacity is adequately maintained.
>
>
> > **Comment4:** (Minor) These are things that might need to be revised in the paper and the appendix:
>
> Thank you very much for pointing out these detailed issues. We sincerely appreciate your careful review. We have carefully addressed and revised these points in the latest version of the paper.
>
>
>
> > **Comment5:** In Figure 3 within the main paper, what do these five bars represent in each corruption type? Please indicate this in a legend within one of the graphs for clarity.
>
>
> Sorry for the confusion. These five bars, transitioning from light to dark, represent the severity levels ranging from 1 to 5 for each corruption type. We have clarified this in the legend for better understanding. Thank you for bringing this to our attention!
>
> > **Comment6:** Regarding analyses conducted in Section 5.1 and Section 5.2, what are the motivations for using the random regrowth procedure for both MEST and GraNet instead of their gradient-based regrowth variants for comparison in Figure 5 and Figure 7? I am asking because the results displayed on the paper (Figure 3, Figure 4, Table 1, and Table 3) use the latter instead of the former.
>
> For the ImageNet dataset, our paper primarily utilizes the gradient-based regrowth approach, **as explained in line 269**. This choice is motivated by its compatibility with PyTorch implementation, where the gradient-based regrowth approach achieves strong performance more quickly on the ImageNet dataset. In fact, in Figure 7, both MEST and GraNet use gradient-based regrowth on ImageNet, as shown in Figures 7 (c) and (f).
>
> For small-scale datasets, both gradient-based and random-based regrowth methods achieve decent performance. Therefore, in Figure 5 and Figure 7, we only included results using the random-based regrowth approach for these datasets. However, to maintain consistency with Figure 3, Figure 4, Table 1, and Table 3, we will add the results using gradient-based regrowth for these datasets in the appendix in our final version. Thank you for pointing this out!

---

> ### Author Response · Authors · 2024-11-24
> **To Reviewer MR2K**
>
> > **Comment7:** What causes each DST method to achieve worse performance in a particular type of corruption (i.e., single_freq (single frequency greyscale) in ImageNet-C-bar)?
>
> From our analysis in Section 5, we find that DST models exhibit **robustness to general high-frequency corruption** by reducing their reliance on high-frequency components, effectively demonstrating a low-frequency bias. This low-frequency bias allows the models to maintain performance under high-frequency corruptions like noise corruption, as these corruptions broadly affect the high-frequency spectrum while leaving low-frequency information largely intact. Here we will briefly to explain why DST method to achieve worse performance.
>
> **1. How SingleFrequencyGreyscale Corruption Works:**
>
> According to the original paper and its implementation (base code: https://github.com/facebookresearch/augmentation-corruption), we can have:
>
> - For each pixel $(x, y)$, a sinusoidal noise value is generated:
>
> $$
> \operatorname{noise}(x, y)=s \times \sin \left(freq_x \cdot x+freq_y \cdot y\right)
> $$
>
> - Here, $freq_x$ and $freq_y$ are randomly chosen spatial frequencies, and $s$ controls the intensity of the corruption.
> - In the frequency domain, SingleFrequencyGreyscale Corruption generates noise at a randomly chosen specific frequency and applies it to each image, creating patterns such as stripes or waves that represent that frequency. Across the entire dataset, **SingleFrequencyGreyscale includes corruptions at various frequencies**.
>
> **2. Impact on DST Models:**
>
> In the whole corrupted test set, the SingleFrequencyGreyscale Corruption introduces a uniform frequencies attentuation for DST models.
>
> While DST models reduce reliance on high-frequency components, they depend on low-frequency information for robust classification. SingleFrequencyGreyscale Corruption disrupts not only high-frequency but also low-frequency information (depending on the chosen $freq_x$ and $freq_y$ ). As analyzed in Section 5.2, particularly in Figure 7 ( c ) and (f), the ImageNet test set demonstrates that under the same attenuation range, **performance drops are significantly more pronounced for low-frequency attenuation** compared to high-frequency attenuation. So such frequencies attentuation in the low-frequency spectrum **dominants** the severe performance degradation in DST models.
>
> Thank you for your feedback on our response. Please let us know if you still have any additional questions or would like further clarification. If you feel that we have addressed all of your concerns, we hope you might reconsider your score. Thanks again for your time and effort.

---

> > ### Comment · Reviewer_MR2K · 2024-11-27
> >
> > Dear Authors,
> >
> > Thank you for answering the concerns I have regarding the paper. I raise my rating from 5 to 6.

---

> > > ### Author Response · Authors · 2024-11-27
> > >
> > > Dear Reviewer MR2K,
> > >
> > > We are pleased to hear that we have addressed your concerns and thank you for your positive support!

---

### Author Response · Authors · 2024-11-24
**To All Reviewers**

We thank the reviewers for their constructive suggestions and in-depth analysis, which is helpful for our work, we truly appreciate it.


We are delighted to note that all reviewers appreciated our **clear presentation** and the **broad range of experiments** across multiple domains, architectures, and DST algorithms. Reviewers commended the **comprehensive evaluations** (MR2K, Khz6, ncJc), the **engaging structure** (EZM6), the **sufficient analysis** (MR2K), and the **valuable insights and investigation from different angles** (ncJc). Additionally, reviewer MR2K acknowledged the paper is **well-organized**, making it easy to read and comprehend, while reviewer ncJc appreciated the **well-explained** preliminaries.


In response to the reviewers' valuable feedback, we have made earnest efforts to address all raised concerns. A summary of our responses is as follows:

- Explained and clarified the **sparsity level settings** used in the experiments (Reviewer MR2K).

- **Provided detailed explanations** and highlighted the differences between our work and related studies (Reviewer EZM6).

- Extended our analysis to include results on additional tasks, such as **Segmentation** (Reviewers ncJc).

- Clarified results in the appendix and expanded our analysis to include evaluations on **other out-of-distribution datasets** (Reviewer Khz6)

- **Added additional experiments** incorporating state-of-the-art (SOTA) methods (Reviewer Khz6).

Should there be any points that remain unclear or require further clarification, please do not hesitate to bring them to our attention. We are open to any additional feedback, comments, or suggestions, and we sincerely appreciate your continued engagement in enhancing the quality of our work.

---

### Meta-Review · Area_Chair_nENY · 2024-12-24

**Metareview:**

The paper investigates the robustness of Dynamic Sparse Training (DST) compared to Dense Training under various image corruption scenarios. It challenges conventional assumptions by demonstrating that DST can consistently outperform dense training in robustness, particularly under moderate sparsity levels (10–50%), without incurring additional resource costs. Through extensive experiments across multiple datasets and models, the paper highlights DST’s inherent robustness advantages, potentially attributed to reduced reliance on high-frequency features.

Reviewer Khz6 maintained a negative stance, primarily citing concerns about baseline accuracy discrepancies and robustness performance under state-of-the-art (SOTA) methods. Specifically, they questioned the clean accuracy gaps between dense and sparse models, arguing that DST’s robustness benefits might come at the expense of clean accuracy. The reviewer also critiqued inconsistencies in the reported results for FAN-Tiny-Hybrid and RVT-Tiny models, claiming that the claimed robustness improvements were undermined by notable drops in clean accuracy. Additionally, Khz6 challenged the significance of robustness gains over existing robust training techniques, expressing skepticism about the novelty and generalizability of the findings.

The authors provided detailed rebuttals, addressing each concern systematically. They clarified that clean accuracy gaps were primarily due to differences in training conditions (e.g., batch size and epoch count) and conducted additional experiments under comparable settings, showing that DST maintained its robustness advantages. For FAN-Tiny-Hybrid and RVT-Tiny, the authors provided updated results and emphasized that while slight clean accuracy drops occurred, they were offset by significant robustness gains, aligning with the paper’s central thesis. Despite these clarifications and added evidence, Khz6 did not update their review or acknowledge the rebuttal, leaving their concerns unresolved.

Considering the paper’s strong empirical contributions and the authors’ rigorous efforts to address reviewer concerns, the AC recommends acceptance. While the disagreement with Khz6 remains unresolved, the rebuttal effectively addressed the key issues, and the findings provide valuable insights into DST’s robustness properties. This work is expected to stimulate further research in robust sparse training methods, making it a meaningful contribution to the field.

**Additional Comments On Reviewer Discussion:**

Please refer to the meta-review.

---

### Decision · Program_Chairs · 2025-01-22

Accept (Poster)